# EXTREME Q-LEARNING: MAXENT RL WITHOUT ENTROPY

**Divyansh Garg**[*]
Stanford University
divgarg@stanford.edu

**Joey Hejna**[*]
Stanford University
jhejna@stanford.edu

**Matthieu Geist**
Google Brain
mfgeist@google.com

**Stefano Ermon**
Stanford University
ermon@stanford.edu

## ABSTRACT

Modern Deep Reinforcement Learning (RL) algorithms require estimates of the maximal Q-value, which are difficult to compute in continuous domains with an infinite number of possible actions. In this work, we introduce a new update rule for online and offline RL which directly models the maximal value using Extreme Value Theory (EVT), drawing inspiration from economics. By doing so, we avoid computing Q-values using out-of-distribution actions which is often a substantial source of error. Our key insight is to introduce an objective that directly estimates the optimal soft-value functions (LogSumExp) in the maximum entropy RL setting without needing to sample from a policy. Using EVT, we derive our *Extreme Q-Learning* framework and consequently online and, for the first time, offline MaxEnt Q-learning algorithms, that do not explicitly require access to a policy or its entropy. Our method obtains consistently strong performance in the D4RL benchmark, outperforming prior works by *10+ points* on the challenging Franka Kitchen tasks while offering moderate improvements over SAC and TD3 on online DM Control tasks. Visualizations and code can be found on our website [1].

## 1 INTRODUCTION

Modern Deep Reinforcement Learning (RL) algorithms have shown broad success in challenging control (Haarnoja et al., 2018; Schulman et al., 2015) and game-playing domains (Mnih et al., 2013). While tabular Q-iteration or value-iteration methods are well understood, state of the art RL algorithms often make theoretical compromises in order to deal with deep networks, high dimensional state spaces, and continuous action spaces. In particular, standard Q-learning algorithms require computing the max or soft-max over the Q-function in order to fit the Bellman equations. Yet, almost all current off-policy RL algorithms for continuous control only indirectly estimate the Q-value of the next state with separate policy networks. Consequently, these methods only estimate the $Q$-function of the current policy, instead of the optimal $Q^*$, and rely on policy improvement via an actor. Moreover, actor-critic approaches on their own have shown to be catastrophic in the offline settings where actions sampled from a policy are consistently out-of-distribution (Kumar et al., 2020; Fujimoto et al., 2018). As such, computing $\max Q$ for Bellman targets remains a core issue in deep RL.

One popular approach is to train Maximum Entropy (MaxEnt) policies, in hopes that they are more robust to modeling and estimation errors (Ziebart, 2010). However, the Bellman backup $\mathcal{B}^*$ used in MaxEnt RL algorithms still requires computing the **log-partition function over Q-values**, *which is usually intractable in high-dimensional action spaces.* Instead, current methods like SAC (Haarnoja et al., 2018) rely on auxiliary policy networks, and as a result do not estimate $\mathcal{B}^*$, the optimal Bellman backup. Our key insight is to apply extreme value analysis used in branches of Finance and Economics to Reinforcement Learning. Ultimately, this will allow us to directly model the LogSumExp over Q-functions in the MaxEnt Framework.

---

[*]Equal Contribution
[1]https://div99.github.io/XQL/

Intuitively, reward or utility-seeking agents will consider the maximum of the set of possible future returns. The Extreme Value Theorem (EVT) tells us that maximal values drawn from any exponential tailed distribution follows the Generalized Extreme Value (GEV) Type-1 distribution, also referred to as the Gumbel Distribution $\mathcal{G}$. The Gumbel distribution is thus a prime candidate for modeling errors in Q-functions. In fact, McFadden's 2000 Nobel-prize winning work in Economics on discrete choice models (McFadden, 1972) showed that soft-optimal utility functions with logit (or softmax) choice probabilities naturally arise when utilities are assumed to have Gumbel-distributed errors. This was subsequently generalized to stochastic MDPs by Rust (1986). Nevertheless, these results have remained largely unknown in the RL community. By introducing a novel loss optimization framework, we bring them into the world of modern deep RL.

Empirically, we find that even modern deep RL approaches, for which errors are typically assumed to be Gaussian, exhibit errors that better approximate the Gumbel Distribution, see Figure 1. By assuming errors to be Gumbel distributed, we obtain Gumbel Regression, a consistent estimator over log-partition functions even in continuous spaces. Furthermore, making this assumption about $Q$-values lets us derive a new Bellman loss objective that directly solves for the optimal MaxEnt Bellman operator $\mathcal{B}^*$, instead of the operator under the current policy $\mathcal{B}^\pi$. As soft optimality emerges from our framework, we can run MaxEnt RL independently of the policy. In the online setting, we avoid using a policy network to explicitly compute entropies. In the offline setting, we completely avoid sampling from learned policy networks, minimizing the aforementioned extrapolation error. Our resulting algorithms surpass or consistently match state-of-the-art (SOTA) methods while being practically simpler.

In this paper we outline the theoretical motivation for using Gumbel distributions in reinforcement learning, and show how it can be used to derive practical online and offline MaxEnt RL algorithms. Concretely, our contributions are as follows:

- We motivate Gumbel Regression and show it allows calculation of the **log-partition function (LogSumExp) in continuous spaces**. We apply it to MDPs to present a novel loss objective for RL using maximum-likelihood estimation.

- Our formulation extends soft-Q learning to offline RL as well as continuous action spaces without the need of policy entropies. It allows us to compute optimal soft-values $V^*$ and soft-Bellman updates $\mathcal{B}^*$ using SGD, which are usually intractable in continuous settings.

- We provide the missing theoretical link between soft and conservative Q-learning, showing how these formulations can be made equivalent. We also show how Max-Ent RL emerges naturally from vanilla RL as a conservatism in our framework.

- Finally, we empirically demonstrate strong results in Offline RL, improving over prior methods by a large margin on the D4RL Franka Kitchen tasks, and performing moderately better than SAC and TD3 in Online RL, while theoretically avoiding actor-critic formulations.

## 2 PRELIMINARIES

In this section we introduce Maximium Entropy (MaxEnt) RL and Extreme Value Theory (EVT), which we use to motivate our framework to estimate extremal values in RL.

We consider an infinite-horizon Markov decision process (MDP), defined by the tuple $(\mathcal{S}, \mathcal{A}, \mathcal{P}, r, \gamma)$, where $\mathcal{S}, \mathcal{A}$ represent state and action spaces, $\mathcal{P}(\mathbf{s}'|\mathbf{s}, \mathbf{a})$ represents the environment dynamics, $r(\mathbf{s}, \mathbf{a})$ represents the reward function, and $\gamma \in (0, 1)$ represents the discount factor. In the offline RL setting, we are given a dataset $\mathcal{D} = (\mathbf{s}, \mathbf{a}, r, \mathbf{s}')$ of tuples sampled from trajectories under a behavior policy $\pi_{\mathcal{D}}$ without any additional environment interactions. We use $\rho_\pi(\mathbf{s})$ to denote the distribution of states that a policy $\pi(\mathbf{a}|\mathbf{s})$ generates. In the MaxEnt framework, an MDP with entropy-regularization is referred to as a soft-MDP (Bloem & Bambos, 2014) and we often use this notation.

### 2.1 MAXIMUM ENTROPY RL

Standard RL seeks to learn a policy that maximizes the expected sum of (discounted) rewards $\mathbb{E}_\pi \left[ \sum_{t=0}^\infty \gamma^t r(\mathbf{s}_t, \mathbf{a}_t) \right]$, for $(\mathbf{s}_t, \mathbf{a}_t)$ drawn at timestep $t$ from the trajectory distribution that $\pi$ generates. We consider a generalized version of Maximum Entropy RL that augments the standard reward objective with the KL-divergence between the policy and a reference distribution $\mu$:

$\mathbb{E}_\pi[\sum_{t=0}^\infty \gamma^t(r(\mathbf{s}_t, \mathbf{a}_t) - \beta \log \frac{\pi(\mathbf{a}_t|\mathbf{s}_t)}{\mu(\mathbf{a}_t|\mathbf{s}_t)})]$, where $\beta$ is the regularization strength. When $\mu$ is uniform $\mathcal{U}$, this becomes the standard MaxEnt objective used in online RL up to a constant. In the offline RL setting, we choose $\mu$ to be the behavior policy $\pi_\mathcal{D}$ that generated the fixed dataset $\mathcal{D}$. Consequently, this objective enforces a conservative KL-constraint on the learned policy, keeping it close to the behavior policy (Neu et al., 2017; Haarnoja et al., 2018).

In MaxEnt RL, the soft-Bellman operator $\mathcal{B}^* : \mathbb{R}^{\mathcal{S} \times \mathcal{A}} \to \mathbb{R}^{\mathcal{S} \times \mathcal{A}}$ is defined as $(\mathcal{B}^* Q)(\mathbf{s}, \mathbf{a}) = r(\mathbf{s}, \mathbf{a}) + \gamma \mathbb{E}_{\mathbf{s}' \sim \mathcal{P}(\cdot|\mathbf{s}, \mathbf{a})} V^*(\mathbf{s}')$ where $Q$ is the soft-Q function and $V^*$ is the optimal soft-value satisfying:

$$V^*(\mathbf{s}) = \beta \log \sum_{\mathbf{a}} \mu(\mathbf{a}|\mathbf{s}) \exp(Q(\mathbf{s}, \mathbf{a})/\beta) := \mathbb{L}^\beta_{a \sim \mu(\cdot|\mathbf{s})}[Q(\mathbf{s}, \mathbf{a})], \tag{1}$$

where we denote the log-sum-exp (LSE) using an operator $\mathbb{L}^\beta$ for succinctness[2]. The soft-Bellman operator has a unique contraction $Q^*$ (Haarnoja et al., 2018) given by the soft-Bellman equation: $Q^* = \mathcal{B}^* Q^*$ and the optimal policy satisfies (Haarnoja et al., 2017):

$$\pi^*(\mathbf{a}|\mathbf{s}) = \mu(\mathbf{a}|\mathbf{s}) \exp((Q^*(\mathbf{s}, \mathbf{a}) - V^*(\mathbf{s}))/\beta). \tag{2}$$

Instead of estimating soft-values for a policy $V^\pi(\mathbf{s}) = \mathbb{E}_{\mathbf{a} \sim \pi(\cdot|\mathbf{s})}\left[Q(\mathbf{s}, \mathbf{a}) - \beta \log \frac{\pi(\mathbf{a}|\mathbf{s})}{\mu(\mathbf{a}|\mathbf{s})}\right]$, our approach will seek to directly fit the optimal soft-values $V^*$, i.e. the log-sum-exp (LSE) of Q values.

## 2.2 EXTREME VALUE THEOREM

The Fisher-Tippett or Extreme Value Theorem tells us that the maximum of i.i.d. samples from exponentially tailed distributions will asymptotically converge to the Gumbel distribution $\mathcal{G}(\mu, \beta)$, which has PDF $p(x) = \exp(-(z + e^{-z}))$ where $z = (x - \mu)/\beta$ with location parameter $\mu$ and scale parameter $\beta$.

**Theorem 1** (Extreme Value Theorem (EVT) (Mood, 1950; Fisher & Tippett, 1928)). *For i.i.d. random variables $X_1, ..., X_n \sim f_X$, with exponential tails, $\lim_{n \to \infty} \max_i(X_i)$ follows the Gumbel (GEV-1) distribution. Furthermore, $\mathcal{G}$ is max-stable, i.e. if $X_i \sim \mathcal{G}$, then $\max_i(X_i) \sim \mathcal{G}$ holds.*

This result is similar to the Central Limit Theorem (CLT), which states that means of i.i.d. errors approach the normal distribution. Thus, under a chain of max operations, any i.i.d. exponential tailed errors[3] will tend to become Gumbel distributed and stay as such. EVT will ultimately suggest us to characterize nested errors in Q-learning as following a Gumbel distribution. In particular, the Gumbel distribution $\mathcal{G}$ exhibits unique properties we will exploit.

One intriguing consequence of the Gumbel's max-stability is its ability to convert the maximum over a discrete set into a softmax. This is known as the *Gumbel-Max Trick* (Papandreou & Yuille, 2010; Hazan & Jaakkola, 2012). Concretely for i.i.d. $\epsilon_i \sim \mathcal{G}(0, \beta)$ added to a set $\{x_1, ..., x_n\} \in \mathbb{R}$, $\max_i(x_i + \epsilon_i) \sim \mathcal{G}(\beta \log \sum_i \exp(x_i/\beta), \beta)$, and $\text{argmax}(x_i + \epsilon_i) \sim \text{softmax}(x_i/\beta)$. Furthermore, the Max-trick is unique to the Gumbel (Luce, 1977). These properties lead into the McFadden-Rust model (McFadden, 1972; Rust, 1986) of MDPs as we state below.

**McFadden-Rust model:** An MDP following the standard Bellman equations with stochasticity in the rewards due to unobserved state variables will satisfy the soft-Bellman equations over the observed state with actual rewards $\bar{r}(\mathbf{s}, \mathbf{a})$, given two conditions:

1. Additive separability (AS): observed rewards have additive i.i.d. Gumbel noise, i.e. $r(\mathbf{s}, \mathbf{a}) = \bar{r}(\mathbf{s}, \mathbf{a}) + \epsilon(\mathbf{s}, \mathbf{a})$, with actual rewards $\bar{r}(\mathbf{s}, \mathbf{a})$ and i.i.d. noise $\epsilon(\mathbf{s}, \mathbf{a}) \sim \mathcal{G}(0, \beta)$.
2. Conditional Independence (CI): the noise $\epsilon(\mathbf{s}, \mathbf{a})$ in a given state-action pair is conditionally independent of that in any other state-action pair.

Moreover, the converse also holds: Any MDP satisfying the Bellman equations and following a $\text{softmax}$ policy, necessarily has any i.i.d. noise in the rewards with $AS + CI$ conditions be Gumbel distributed. These results were first shown to hold in discrete choice theory by McFadden (1972), with the $AS + CI$ conditions derived by Rust (1986) for discrete MDPs. We formalize these results in Appendix A and give succinct proofs using the developed properties of the Gumbel distribution. These results enable the view of a soft-MDP as an MDP with hidden i.i.d. Gumbel noise in the rewards. Notably, this result gives a different interpretation of a soft-MDP than entropy regularization to allow us to recover the soft-Bellman equations.

---

[2]In continuous action spaces, the sum over actions is replaced with an integral over the distribution $\mu$.

[3]Bounded random variables are sub-Gaussian (Young, 2020) which have exponential tails.

# 3 EXTREME Q-LEARNING

In this section, we motivate our Extreme Q-learning framework, which directly models the soft-optimal values $V^*$, and show it naturally extends soft-Q learning. Notably, we use the Gumbel distribution to derive a new optimization framework for RL via maximum-likelihood estimation and apply it to both online and offline settings.

## 3.1 GUMBEL ERROR MODEL

Although assuming Gumbel errors in MDPs leads to intriguing properties, it is not obvious why the errors might be distributed as such. First, we empirically investigate the distribution of Bellman errors by computing them over the course of training. Specifically, we compute $r(\mathbf{s}, \mathbf{a}) - \gamma Q(\mathbf{s}', \pi(\mathbf{s}')) - Q(\mathbf{s}, \mathbf{a})$ for samples $(\mathbf{s}, \mathbf{a}, \mathbf{s}')$ from the replay-buffer using a single $Q$-function from SAC (Haarnoja et al., 2018) (See Appendix D for more details). In Figure 1, we find the errors to be skewed and better fit by a Gumbel distribution. We explain this using EVT.

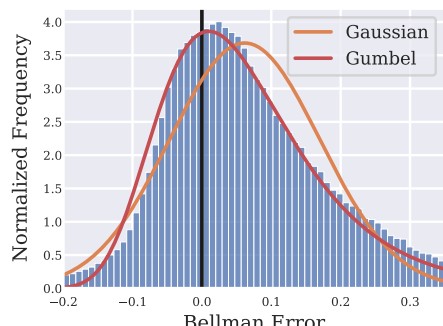

Figure 1: Bellman errors from SAC on Cheetah-Run (Tassa et al., 2018). The Gumbel distribution better captures the skew versus the Gaussian. Plots for TD3 and more environments can be found in Appendix D.

Consider fitting $Q$-functions by learning an unbiased function approximator $\hat{Q}$ to solve the Bellman equation. We will assume access to $M$ such function approximators, each of which are assumed to be independent e.g. parallel runs of a model over an experiment. We can see approximate Q-iteration as performing:

$$\hat{Q}_t(\mathbf{s}, \mathbf{a}) = \bar{Q}_t(\mathbf{s}, \mathbf{a}) + \epsilon_t(\mathbf{s}, \mathbf{a}), \tag{3}$$

where $\mathbb{E}[\hat{Q}] = \bar{Q}_t$ is the expected value of our prediction $\hat{Q}_t$ for an intended target $\bar{Q}_t$ over our estimators, and $\epsilon_t$ is the (zero-centered) error in our estimate. Here, we assume the error $\epsilon_t$ comes from the same underlying distribution for each of our estimators, and thus are i.i.d. random variables with a zero-mean. Now, consider the bootstrapped estimate using one of our M estimators chosen randomly:

$$\hat{\mathcal{B}}^* \hat{Q}_t(\mathbf{s}, \mathbf{a}) = r(\mathbf{s}, \mathbf{a}) + \gamma \max_{\mathbf{a}'} \hat{Q}_t(\mathbf{s}', \mathbf{a}') = r(\mathbf{s}, \mathbf{a}) + \gamma \max_{\mathbf{a}'} (\bar{Q}_t(\mathbf{s}', \mathbf{a}') + \epsilon_t(\mathbf{s}', \mathbf{a}')). \tag{4}$$

We now examine what happens after a subsequent update. At time $t + 1$, suppose that we fit a fresh set of $M$ independent functional approximators $\hat{Q}_{t+1}$ with the target $\hat{\mathcal{B}}^* \hat{Q}_t$, introducing a new unbiased error $\epsilon_{t+1}$. Then, for $\bar{Q}_{t+1} = \mathbb{E}[\hat{Q}_{t+1}]$ it holds that

$$\bar{Q}_{t+1}(\mathbf{s}, \mathbf{a}) = r(\mathbf{s}, \mathbf{a}) + \gamma \mathbb{E}_{\mathbf{s}'|\mathbf{s}, \mathbf{a}} [\mathbb{E}_{\epsilon_t}[\max_{\mathbf{a}'} (\bar{Q}_t(\mathbf{s}', \mathbf{a}') + \epsilon_t(\mathbf{s}', \mathbf{a}'))]]. \tag{5}$$

As $\bar{Q}_{t+1}$ is an expectation over both the dynamics and the functional errors, it accounts for all uncertainty (here $\mathbb{E}[\epsilon_{t+1}] = 0$). But, the i.i.d. error $\epsilon_t$ remains and will be propagated through the Bellman equations and its chain of max operations. Due to Theorem 1, $\epsilon_t$ will become Gumbel distributed in the limit of $t$, and remain so due to the Gumbel distribution's max-stability.[4]

This highlights a fundamental issue with approximation-based RL algorithms that minimize the Mean-Squared Error (MSE) in the Bellman Equation: they implicitly assume, via maximum likelihood estimation, that errors are Gaussian. In Appendix A, we further study the propagation of errors using the McFadden-Rust MDP model, and use it to develop a simplified Gumbel Error Model (GEM) for errors under functional approximation. In practice, the Gumbel nature of the errors may be weakened as estimators between timesteps share parameters and errors will be correlated across states and actions.

## 3.2 GUMBEL REGRESSION

The goal of our work is to directly model the log-partition function (LogSumExp) over $Q(s, a)$ to avoid all of the aforementioned issues with taking a max in the function approximation domain.

---

[4]The same holds for soft-MDPs as log-sum-exp can be expanded as a max over i.i.d. Gumbel random vars.

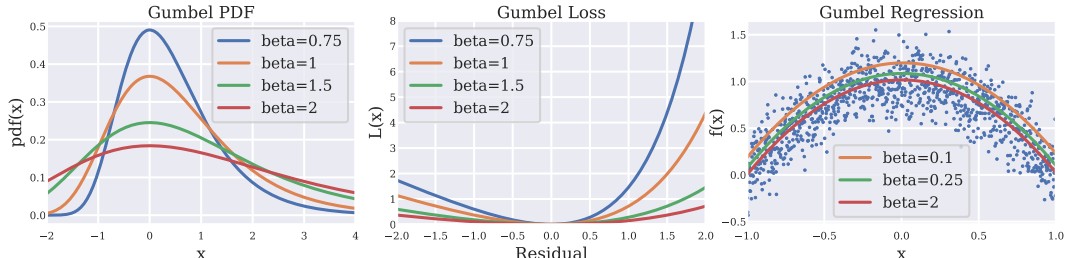

Figure 2: **Left**: The pdf of the Gumbel distribution with $\mu = 0$ and different values of $\beta$. **Center**: Our Gumbel loss for different values of $\beta$. **Right**: Gumbel regression applied to a two-dimensional random variable for different values of $\beta$. The smaller the value of $\beta$, the more the regression fits the extrema.

In this section we derive an objective function that models the LogSumExp by simply assuming errors follow a Gumbel distribution. Consider estimating a parameter $h$ for a random variable $X$ using samples $x_i$ from a dataset $\mathcal{D}$, which have Gumbel distributed noise, i.e. $x_i = h + \epsilon_i$ where $\epsilon_i \sim -\mathcal{G}(0, \beta)$. Then, the average log-likelihood of the dataset $\mathcal{D}$ as a function of $h$ is given as:

$$\mathbb{E}_{x_i \sim \mathcal{D}}\left[\log p(x_i)\right] = \mathbb{E}_{x_i \sim \mathcal{D}}\left[-e^{((x_i - h)/\beta)} + (x_i - h)/\beta\right] \qquad (6)$$

Maximizing the log-likelihood yields the following convex minimization objective in $h$,

$$\mathcal{L}(h) = \mathbb{E}_{x_i \sim \mathcal{D}}\left[e^{(x_i - h)/\beta} - (x_i - h)/\beta - 1\right] \qquad (7)$$

which forms our objective function $\mathcal{L}(\cdot)$, which resembles the Linex loss from econometrics (Parsian & Kirmani, 2002) [5]. $\beta$ is fixed as a hyper-parameter, and we show its affect on the loss in Figure 2. Critically, the minima of this objective under a fixed $\beta$ is given by $h = \beta \log \mathbb{E}_{x_i \sim \mathcal{D}}[e^{x_i/\beta}]$, which resembles the LogSumExp with the summation replaced with an (empirical) expectation. In fact, this solution is the the same as the operator $\mathbb{L}_\mu^\beta(X)$ defined for MaxEnt in Section 2.1 with $x_i$ sampled from $\mu$. In Figure 2, we show plots of Gumbel Regression on a simple dataset with different values of $\beta$. As this objective recovers $\mathbb{L}^\beta(X)$, we next use it to model soft-values in Max-Ent RL.

### 3.2.1 THEORY

Here we show that Gumbel regression is well behaved, considering the previously defined operator $\mathbb{L}^\beta$ for random variables $\mathbb{L}^\beta(X) := \beta \log \mathbb{E}\left[e^{X/\beta}\right]$. First, we show it models the extremum.

**Lemma 3.1.** *For any $\beta_1 > \beta_2$, we have $\mathbb{L}^{\beta_1}(X) < \mathbb{L}^{\beta_2}(X)$. And $\mathbb{L}^\infty(X) = \mathbb{E}[X]$, $\mathbb{L}^0(X) = sup(X)$. Thus, for any $\beta \in (0, \infty)$, the operator $\mathbb{L}^\beta(X)$ is a measure that interpolates between the expectation and the max of $X$.*

The operator $\mathbb{L}^\beta(X)$ is known as the cumulant-generating function or the log-Laplace transform, and is a measure of the tail-risk closely linked to the entropic value at risk (EVaR) (Ahmadi-Javid, 2012) .

**Lemma 3.2.** *The risk measure $\mathcal{L}$ has a unique minima at $\beta \log \mathbb{E}\left[e^{X/\beta}\right]$. And an empirical risk $\hat{\mathcal{L}}$ is an unbiased estimate of the true risk. Furthermore, for $\beta \gg 1$, $\mathcal{L}(\theta) \approx \frac{1}{2\beta^2}\mathbb{E}_{x_i \sim \mathcal{D}}[(x_i - \theta)^2]$, thus behaving as the MSE loss with errors $\sim \mathcal{N}(0, \beta)$.*

In particular, the empirical loss $\hat{\mathcal{L}}$ over a dataset of $N$ samples can be minimized using stochastic gradient-descent (SGD) methods to give an unbiased estimate of the LogSumExp over the $N$ samples.

**Lemma 3.3.** *$\hat{\mathbb{L}}^\beta(X)$ over a finite $N$ samples is a consistent estimator of the log-partition function $\mathbb{L}^\beta(X)$. Similarly, $\exp(\hat{\mathbb{L}}^\beta(X)/\beta)$ is an unbiased estimator for the partition function $Z = \mathbb{E}\left[e^{X/\beta}\right]$*

We provide PAC learning bounds for Lemma 3.3, and further theoretical discussion on Gumbel Regression in Appendix B.

### 3.3 MAXENT RL WITHOUT ENTROPY

Given Gumbel Regression can be used to directly model the LogSumExp , we apply it to Q-learning. First, we connect our framework to conservative Q-learning (Kumar et al., 2020).

---

[5]We add $-1$ to make the loss 0 for a perfect fit, as $e^x - x - 1 \geq 0$ with equality at $x = 0$.

**Lemma 3.4.** *Consider the loss objective over Q-functions:*

$$\mathcal{L}(Q) = \mathbb{E}_{\mathbf{s} \sim \rho_\mu, \mathbf{a} \sim \mu(\cdot|\mathbf{s})} \left[ e^{(\mathcal{T}^\pi \hat{Q}^k(\mathbf{s},\mathbf{a}) - Q(\mathbf{s},\mathbf{a}))/\beta} \right] - \mathbb{E}_{\mathbf{s} \sim \rho_\mu, \mathbf{a} \sim \mu(\cdot|\mathbf{s})}[(\mathcal{T}^\pi \hat{Q}^k(\mathbf{s},\mathbf{a}) - Q(\mathbf{s},\mathbf{a}))/\beta] - 1 \tag{8}$$

*where $\mathcal{T}^\pi := r(\mathbf{s},\mathbf{a}) + \gamma \mathbb{E}_{\mathbf{s}'|\mathbf{s},\mathbf{a}} \mathbb{E}_{\mathbf{a}' \sim \pi}[Q(\mathbf{s}',\mathbf{a}')]$ is the vanilla Bellman operator under the policy $\pi(\mathbf{a}|\mathbf{s})$. Then minimizing $\mathcal{L}$ gives the update rule:*

$$\forall \mathbf{s},\mathbf{a},k \;\; \hat{Q}^{k+1}(\mathbf{s},\mathbf{a}) = \mathcal{T}^\pi \hat{Q}^k(\mathbf{s},\mathbf{a}) - \beta \log \frac{\pi(\mathbf{a} \mid \mathbf{s})}{\mu(\mathbf{a} \mid \mathbf{s})} = \mathcal{B}^\pi \hat{Q}^k(\mathbf{s},\mathbf{a}).$$

The above lemma transforms the regular Bellman backup into the soft-Bellman backup without the need for entropies, letting us convert standard RL into MaxEnt RL. Here, $\mathcal{L}(\cdot)$ does a conservative Q-update similar to CQL (Kumar et al., 2020) with the nice property that the implied conservative term is just the KL-constraint between $\pi$ and $\mu$.[6] This enforces a entropy-regularization on our policy with respect to the behavior policy without the need of entropy. Thus, soft-Q learning naturally emerges as a conservative update on regular Q-learning under our objective. Here, Equation 8 is the dual of the KL-divergence between $\mu$ and $\pi$ (Garg et al., 2021), and we motivate this objective for RL and establish formal equivalence with conservative Q-learning in Appendix C.

In our framework, we use the MaxEnt Bellman operator $\mathcal{B}^*$ which gives our *ExtremeQ* loss, which is the same as our Gumbel loss from the previous section:

$$\mathcal{L}(Q) = \mathbb{E}_{\mathbf{s},\mathbf{a} \sim \mu} \left[ e^{(\hat{\mathcal{B}}^* \hat{Q}^k(\mathbf{s},\mathbf{a}) - Q(\mathbf{s},\mathbf{a}))/\beta} \right] - \mathbb{E}_{\mathbf{s},\mathbf{a} \sim \mu}[(\hat{\mathcal{B}}^* \hat{Q}^k(\mathbf{s},\mathbf{a}) - Q(\mathbf{s},\mathbf{a}))/\beta] - 1 \tag{9}$$

This gives an update rule: $\hat{Q}^{k+1}(\mathbf{s},\mathbf{a}) = \mathcal{B}^* \hat{Q}^k(\mathbf{s},\mathbf{a})$. $\mathcal{L}(\cdot)$ here requires estimation of $\mathcal{B}^*$ which is very hard in continuous action spaces. Under deterministic dynamics, $\mathcal{L}$ can be obtained without $\mathcal{B}^*$ as shown in Appendix C. However, in general we still need to estimate $\mathcal{B}^*$. Next, we motivate how we can solve this issue. Consider the soft-Bellman equation from Section 2.1 (Equation 1),

$$\mathcal{B}^* Q = r(\mathbf{s},\mathbf{a}) + \gamma \mathbb{E}_{\mathbf{s}' \sim P(\cdot|\mathbf{s},\mathbf{a})}[V^*(\mathbf{s}')], \tag{10}$$

where $V^*(\mathbf{s}) = \mathbb{L}^\beta_{\mathbf{a} \sim \mu(\cdot|\mathbf{s}')}[Q(\mathbf{s},\mathbf{a})]$. Then $V^*$ can be directly estimated using Gumbel regression by setting the temperature $\beta$ to the regularization strength in the MaxEnt framework. This gives us the following *ExtremeV* loss objective:

$$\mathcal{J}(V) = \mathbb{E}_{\mathbf{s},\mathbf{a} \sim \mu} \left[ e^{(\hat{Q}^k(\mathbf{s},\mathbf{a}) - V(\mathbf{s}))/\beta} \right] - \mathbb{E}_{\mathbf{s},\mathbf{a} \sim \mu}[(\hat{Q}^k(\mathbf{s},\mathbf{a}) - V(\mathbf{s}))/\beta] - 1. \tag{11}$$

**Lemma 3.5.** *Minimizing $\mathcal{J}$ over values gives the update rule: $\hat{V}^k(\mathbf{s}) = \mathbb{L}^\beta_{\mathbf{a} \sim \mu(\cdot|\mathbf{s})}[\hat{Q}^k(\mathbf{s},\mathbf{a})]$.*

Then we can obtain $V^*$ from $Q(s,a)$ using Gumbel regression and substitute in Equation 10 to estimate the optimal bellman backup $\mathcal{B}^* Q$. Thus, Lemma 3.4 and 3.5 give us a scheme to solve the Max-Ent RL problem without the need of entropy.

## 3.4 LEARNING POLICIES

In the above section we derived a $Q$-learning strategy that does not require explicit use of a policy $\pi$. However, in continuous settings we still often want to recover a policy that can be run in the environment. Per Eq. 2 (Section 2.2), the optimal MaxEnt policy $\pi^*(\mathbf{a}|\mathbf{s}) = \mu(\mathbf{a}|\mathbf{s})e^{(Q(\mathbf{s},\mathbf{a}) - V(\mathbf{s}))/\beta}$. By minimizing the forward KL-divergence between $\pi$ and the optimal $\pi^*$ induced by $Q$ and $V$ we obtain the following training objective:

$$\pi^* = \operatorname*{argmax}_\pi \mathbb{E}_{\rho_\mu(\mathbf{s},\mathbf{a})}[e^{(Q(\mathbf{s},\mathbf{a}) - V(\mathbf{s}))/\beta} \log \pi]. \tag{12}$$

If we take $\rho_\mu$ to be a dataset $\mathcal{D}$ generated from a behavior policy $\pi_\mathcal{D}$, we exactly recover the AWR objective used by prior works in Offline RL (Peng et al., 2019; Nair et al., 2020), which can easily be computed using the offline dataset. This objective does not require sampling actions, which may

---

[6]In fact, theorems of CQL (Kumar et al., 2020) hold for our objective by replacing $D_{CQL}$ with $D_{KL}$.

potentially take $Q(s,a)$ out of distribution. Alternatively, if we want to sample from the policy instead of the reference distribution $\mu$, we can minimize the Reverse-KL divergence which gives us the SAC-like actor update:

$$\pi^* = \underset{\pi}{\arg\max}\, \mathbb{E}_{\rho_\pi(\mathbf{s})\pi(\mathbf{a}|\mathbf{s})}[Q(\mathbf{s},\mathbf{a}) - \beta \log(\pi(\mathbf{a}|\mathbf{s})/\mu(\mathbf{a}|\mathbf{s}))]. \tag{13}$$

Interestingly, we note this doesn't depend on $V(s)$. If $\mu$ is chosen to be the last policy $\pi_k$, the second term becomes the KL-divergence between the current policy and $\pi_k$, performing a trust region update on $\pi$ (Schulman et al., 2015; Vieillard et al., 2020).[7] While estimating the log ratio $\log(\pi(\mathbf{a}|\mathbf{s})/\mu(\mathbf{a}|\mathbf{s}))$ can be difficult depending on choice of $\mu$, our Gumbel Loss $\mathcal{J}$ removes the need for $\mu$ during $Q$ learning by estimating soft-$Q$ values of the form $Q(\mathbf{s},\mathbf{a}) - \beta \log(\pi(\mathbf{a}|\mathbf{s})/\mu(\mathbf{a}|\mathbf{s}))$.

## 3.5 PRACTICAL ALGORITHMS

In this section we develop a practical approach to Extreme Q-learning ($\mathcal{X}$-QL) for both online and offline RL. We consider parameterized functions $V_\theta(\mathbf{s})$, $Q_\phi(\mathbf{s},\mathbf{a})$, and $\pi_\psi(\mathbf{a}|\mathbf{s})$ and let $\mathcal{D}$ be the training data distribution. A core issue with directly optimizing Eq. 10 is over-optimism about dynamics (Levine, 2018) when using simple-sample estimates for the Bellman backup. To overcome this issue in stochastic settings, we separate out the optimization of $V_\theta$ from that of $Q_\phi$ following Section 3.3. We learn $V_\theta$ using Eq. 11 to directly fit the optimal soft-values $V^*(\mathbf{s})$ based on Gumbel regression. Using $V_\theta(\mathbf{s}')$ we can

---
**Algorithm 1** Extreme Q-learning ($\mathcal{X}$-QL) (Under Stochastic Dynamics)

---
1: Init $Q_\phi$, $V_\theta$, and $\pi_\psi$
2: Let $\mathcal{D} = \{(\mathbf{s},\mathbf{a},r,\mathbf{s}')\}$ be data from $\pi_\mathcal{D}$ (offline) or replay buffer (online)
3: **for** step $t$ in $\{1...N\}$ **do**
4:     Train $Q_\phi$ using $\mathcal{L}(\phi)$ from Eq. 14
5:     Train $V_\theta$ using $\mathcal{J}(\theta)$ from Eq. 11
       (with $\mathbf{a} \sim \mathcal{D}$ (offline) or $\mathbf{a} \sim \pi_\psi$ (online))
6:     Update $\pi_\psi$ via Eq. 12 (offline) or Eq. 13 (online)
7: **end for**

---

get single-sample estimates of $\mathcal{B}^*$ as $r(\mathbf{s},\mathbf{a}) + \gamma V_\theta(\mathbf{s}')$. Now we can learn an unbiased expectation over the dynamics, $Q_\phi \approx \mathbb{E}_{\mathbf{s}'|\mathbf{s},\mathbf{a}}[r(\mathbf{s},\mathbf{a}) + \gamma V_\theta(\mathbf{s}')]$ by minimizing the Mean-squared-error (MSE) loss between the single-sample targets and $Q_\phi$:

$$\mathcal{L}(\phi) = \mathbb{E}_{(\mathbf{s},\mathbf{a},\mathbf{s}')\sim\mathcal{D}}\left[(Q_\phi(\mathbf{s},\mathbf{a}) - r(\mathbf{s},\mathbf{a}) - \gamma V_\theta(\mathbf{s}'))^2\right]. \tag{14}$$

In deterministic dynamics, our approach is largely simplified and we directly learn a single $Q_\phi$ using Eq. 9 without needing to learn $\mathcal{B}^*$ or $V^*$. Similarly, we learn soft-optimal policies using Eq. 12 (offline) or Eq. 13 (online) settings.

**Offline RL**. In the offline setting, $\mathcal{D}$ is specified as an offline dataset assumed to be collected with the behavior policy $\pi_\mathcal{D}$. Here, learning values with Eq. 11 has a number of practical benefits. First, we are able to fit the optimal soft-values $V^*$ *without sampling from a policy network*, which has been shown to cause large out-of-distribution errors in the offline setting where mistakes cannot be corrected by collecting additional data. Second, we inherently enforce a KL-constraint on the optimal policy $\pi^*$ and the behavior policy $\pi_\mathcal{D}$. This provides tunable conservatism via the temperature $\beta$. After offline training of $Q_\phi$ and $V_\theta$, we can recover the policy post-training using the AWR objective (Eq. 12). Our practical implementation follows the training style of Kostrikov et al. (2021), but we train value network using using our ExtremeQ loss.

**Online RL**. In the online setting, $\mathcal{D}$ is usually given as a replay buffer of previously sampled states and actions. In practice, however, obtaining a good estimate of $V^*(\mathbf{s}')$ requires that we sample actions with high Q-values instead of uniform sampling from $\mathcal{D}$. As online learning allows agents to correct over-optimism by collecting additional data, we use a previous version of the policy network $\pi_\psi$ to sample actions for the Bellman backup, amounting to the trust-region policy updates detailed at the end of Section 3.4. In practice, we modify SAC and TD3 with our formulation. To embue SAC (Haarnoja et al., 2018) with the benefits of Extreme Q-learning, we simply train $V_\theta$ using Eq. 11 with $\mathbf{s} \sim \mathcal{D}, \mathbf{a} \sim \pi_{\psi_k}(\mathbf{a}|\mathbf{s})$. This means that we do not use action probabilities when updating the value networks, unlike other MaxEnt RL approaches. The policy is learned via the objective $\max_\psi \mathbb{E}[Q_\phi(s, \pi_\psi(s))]$ with added entropy regularization, as SAC does not use a fixed noise schedule. TD3 by default does not use a value network, and thus we use our algorithm for deterministic dynamics by changing the loss to train $Q$ in TD3 to directly follow Eq. 9. The policy is learned as in SAC, except without entropy regularization as TD3 uses a fixed noise schedule.

---
[7]Choosing $\mu$ to be uniform $\mathcal{U}$ gives the regular SAC update.

## 4 EXPERIMENTS

We compare our Extreme Q-Learning ($\mathcal{X}$-QL) approach to state-of-the-art algorithms across a wide set of continuous control tasks in both online and offline settings. In practice, the exponential nature of the Gumbel regression poses difficult optimization challenges. We provide Offline results on Android, details of loss implementation, ablations, and hyperparameters in Appendix D.

### 4.1 OFFLINE RL

Table 1: Averaged normalized scores on MuJoCo locomotion and Ant Maze tasks. $\mathcal{X}$-QL-C gives results with the same consistent hyper-parameters in each domain, and $\mathcal{X}$-QL-T gives results with per-environment $\beta$ and hyper-parameter tuning.

| | Dataset | BC | 10%BC | DT | AWAC | Onestep RL | TD3+BC | CQL | IQL | $\mathcal{X}$-QL C | $\mathcal{X}$-QL T |
|---|---|---|---|---|---|---|---|---|---|---|---|
| Gym | halfcheetah-medium-v2 | 42.6 | 42.5 | 42.6 | 43.5 | **48.4** | 48.3 | 44.0 | 47.4 | 47.7 | 48.3 |
| | hopper-medium-v2 | 52.9 | 56.9 | 67.6 | 57.0 | 59.6 | 59.3 | 58.5 | 66.3 | 71.1 | 74.2 |
| | walker2d-medium-v2 | 75.3 | 75.0 | 74.0 | 72.4 | **81.8** | 83.7 | 72.5 | 78.3 | 81.5 | 84.2 |
| | halfcheetah-medium-replay-v2 | 36.6 | 40.6 | 36.6 | 40.5 | 38.1 | **44.6** | 45.5 | 44.2 | 44.8 | 45.2 |
| | hopper-medium-replay-v2 | 18.1 | 75.9 | 82.7 | 37.2 | **97.5** | 60.9 | 95.0 | 94.7 | 97.3 | 100.7 |
| | walker2d-medium-replay-v2 | 26.0 | 62.5 | 66.6 | 27.0 | 49.5 | **81.8** | 77.2 | 73.9 | 75.9 | 82.2 |
| | halfcheetah-medium-expert-v2 | 55.2 | 92.9 | 86.8 | 42.8 | **93.4** | 90.7 | 91.6 | 86.7 | 89.8 | 94.2 |
| | hopper-medium-expert-v2 | 52.5 | **110.9** | 107.6 | 55.8 | 103.3 | 98.0 | 105.4 | 91.5 | 107.1 | 111.2 |
| | walker2d-medium-expert-v2 | 107.5 | **109.0** | 108.1 | 74.5 | **113.0** | 110.1 | 108.8 | 109.6 | 110.1 | 112.7 |
| AntMaze | antmaze-umaze-v0 | 54.6 | 62.8 | 59.2 | 56.7 | 64.3 | 78.6 | 74.0 | 87.5 | 87.2 | 93.8 |
| | antmaze-umaze-diverse-v0 | 45.6 | 50.2 | 53.0 | 49.3 | 60.7 | 71.4 | **84.0** | 62.2 | 69.17 | 82.0 |
| | antmaze-medium-play-v0 | 0.0 | 5.4 | 0.0 | 0.0 | 0.3 | 10.6 | 61.2 | 71.2 | **73.5** | 76.0 |
| | antmaze-medium-diverse-v0 | 0.0 | 9.8 | 0.0 | 0.7 | 0.0 | 3.0 | 53.7 | **70.0** | 67.8 | 73.6 |
| | antmaze-large-play-v0 | 0.0 | 0.0 | 0.0 | 0.0 | 0.0 | 0.2 | 15.8 | 39.6 | 41 | 46.5 |
| | antmaze-large-diverse-v0 | 0.0 | 6.0 | 0.0 | 1.0 | 0.0 | 0.0 | 14.9 | **47.5** | 47.3 | 49.0 |
| Franka | kitchen-complete-v0 | 65.0 | - | - | - | - | - | 43.8 | 62.5 | 72.5 | 82.4 |
| | kitchen-partial-v0 | 38.0 | - | - | - | - | - | 49.8 | 46.3 | **73.8** | 73.7 |
| | kitchen-mixed-v0 | 51.5 | - | - | - | - | - | 51.0 | 51.0 | **54.6** | 62.5 |
| | runtime | 10m | 10m | 960m | 20m | 20m | 20m | 80m | 20m | 10-20m | 10-20m[*] |

[*]We see very fast convergence for our method on some tasks, and saturate performance at half the iterations as IQL.

Our offline results with fixed hyperparameters for each domain outperform prior methods (Chen et al., 2021; Kumar et al., 2019; 2020; Kostrikov et al., 2021; Fujimoto & Gu, 2021) in several environments, reaching *state-of-the-art* on the Franka Kitchen tasks, as shown in Table 1. We find performance on the Gym locomotion tasks to be already largely saturated without introducing ensembles An et al. (2021), but our method achieves consistently high performance across environments. While we attain good performance using fixed hyper-parameters per domain, $\mathcal{X}$-QL achieves even higher absolute performance and faster convergence than IQL's reported results when hyper-parameters are turned per environment. With additional tuning, we also see particularly large improvements on the AntMaze tasks, which require a significant amount of "stitching" between trajectories (Kostrikov et al., 2021). Full learning curves are in the Appendix. Like IQL, $\mathcal{X}$-QL can be easily fine-tuned using online data to attain even higher performance as shown in Table 2.

### 4.2 ONLINE RL

We compare ExtremeQ variants of SAC (Haarnoja et al., 2018) and TD3 (Fujimoto et al., 2018), denoted $\mathcal{X}$-SAC and $\mathcal{X}$-TD3, to their vanilla versions on tasks in the DM Control, shown in Figure 3. Across all tasks an ExtremeQ variant matches or

Table 2: Finetuning results on the AntMaze environments

| Dataset | CQL | | IQL | | $\mathcal{X}$-QL T | |
|---|---|---|---|---|---|---|
| umaze-v0 | 70.1 | $\rightarrow$ **99.4** | 86.7 | $\rightarrow$ 96.0 | **93.8** | $\rightarrow$ **99.6** |
| umaze-diverse-v0 | 31.1 | $\rightarrow$ **99.4** | 75.0 | $\rightarrow$ 84.0 | **82.0** | $\rightarrow$ 99.0 |
| medium-play-v0 | 23.0 | $\rightarrow$ 0.0 | 72.0 | $\rightarrow$ 95.0 | **76.0** | $\rightarrow$ 97.0 |
| medium-diverse-v0 | 23.0 | $\rightarrow$ 32.3 | 68.3 | $\rightarrow$ 92.0 | **73.6** | $\rightarrow$ **97.1** |
| large-play-v0 | 1.0 | $\rightarrow$ 0.0 | 25.5 | $\rightarrow$ 46.0 | **45.1** | $\rightarrow$ **59.3** |
| large-diverse-v0 | 1.0 | $\rightarrow$ 0.0 | 42.6 | $\rightarrow$ 60.7 | **49.0** | $\rightarrow$ **82.1** |

surpasses the performance of baselines. We see particularly large gains in the Hopper environment, and more significant gains in comparison to TD3 overall. Consistent with SAC (Haarnoja et al., 2018), we find the temperature $\beta$ needs to be tuned for different environments with different reward scales and sparsity. A core component of TD3 introduced by Fujimoto et al. (2018) is Double Q-Learning, which takes the minimum of two $Q$ functions to remove overestimate bias in the Q-target. As we assume errors to be Gumbel distributed, we expect our $\mathcal{X}$-variants to be more robust to such errors. In all environments except Cheetah Run, our $\mathcal{X}$-TD3 without the Double-Q trick, denoted $\mathcal{X}$-QL - DQ, performs better than standard TD3. While the gains from Extreme-Q learning are modest in online settings, none of our methods require access to the policy distribution to learn the Q-values.

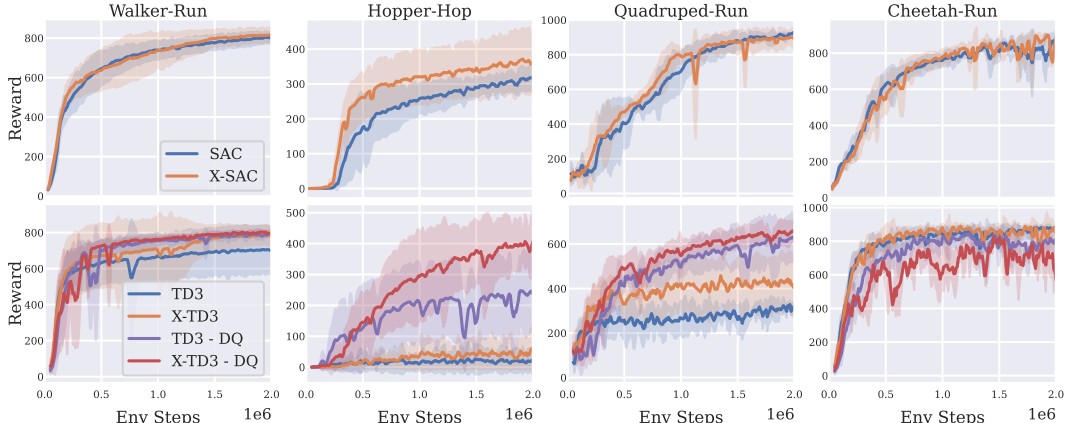

Figure 3: Results on the DM Control for SAC and TD3 based versions of Extreme Q Learning.

## 5 RELATED WORK

Our approach builds on works online and offline RL. Here we review the most salient ones. Inspiration for our framework comes from econometrics (Rust, 1986; McFadden, 1972), and our Gumbel loss is motivated by IQ-Learn (Garg et al., 2021).

**Online RL**. Our work bridges the theoretical gap between RL and Max-Ent RL by introducing our Gumbel loss function. Unlike past work in MaxEnt RL (Haarnoja et al., 2018; Eysenbach & Levine, 2020), our method does not require explicit entropy estimation and instead addresses the problem of obtaining soft-value estimates (LogSumExp) in high-dimensional or continuous spaces (Vieillard et al., 2021) by directly modeling them via our proposed Gumbel loss, which to our knowledge has not previously been used in RL. Our loss objective is intrinsically linked to the KL divergence, and similar objectives have been used for mutual information estimation (Poole et al., 2019) and statistical learning Parsian & Kirmani (2002); Atiyah et al. (2020). IQ-Learn (Garg et al., 2021) proposes learning Q-functions to solve imitation introduced the same loss in IL to obtain an unbiased dual form for the reverse KL-divergence between an expert and policy distribution. Other works have also used forward KL-divergence to derive policy objectives (Peng et al., 2019) or for regularization (Schulman et al., 2015; Abdolmaleki et al., 2018). Prior work in RL has also examined using other types of loss functions (Bas-Serrano et al., 2021) or other formulations of the argmax in order to ease optimization (Asadi & Littman, 2017). Distinct from most off-Policy RL Methods (Lillicrap et al., 2015; Fujimoto et al., 2018; Haarnoja et al., 2018), we directly model $\mathcal{B}^*$ like Haarnoja et al. (2017); Heess et al. (2015) but attain significantly more stable results.

**Offline RL**. Prior works in offline RL can largely be categorized as relying on constrained or regularized Q-learning (Wu et al., 2019; Fujimoto & Gu, 2021; Fujimoto et al., 2019; Kumar et al., 2019; 2020; Nair et al., 2020), or extracting a greedy policy from the known behavior policy (Peng et al., 2019; Brandfonbrener et al., 2021; Chen et al., 2021). Most similar to our work, IQL (Kostrikov et al., 2021) fits expectiles of the Q-function of the behavior policy, but is not motivated to solve a particular problem or remain conservative. On the other hand, conservatism in CQL (Kumar et al., 2020) is motivated by lower-bounding the Q-function. Our method shares the best of both worlds – like IQL we do not evaluate the Q-function on out of distribution actions and like CQL we enjoy the benefits of conservatism. Compared to CQL, our approach uses a KL constraint with the behavior policy, and for the first time extends soft-Q learning to offline RL without needing a policy or explicit entropy values. Our choice of using the reverse KL divergence for offline RL follows closely with BRAC (Wu et al., 2019) but avoids learning a policy during training.

## 6 CONCLUSION

We propose Extreme Q-Learning, a new framework for MaxEnt RL that directly estimates the optimal Bellman backup $\mathcal{B}^*$ without relying on explicit access to a policy. Theoretically, we bridge the gap between the regular, soft, and conservative Q-learning formulations. Empirically, we show that our framework can be used to develop simple, performant RL algorithms. A number of future directions remain such as improving stability with training with the exponential Gumbel Loss function and integrating automatic tuning methods for temperature $\beta$ like SAC (Haarnoja et al., 2018). Finally, we hope that our framework can find general use in Machine Learning for estimating log-partition functions.

**Acknowledgements**

Div derived the theory for Extreme Q-learning and Gumbel regression framework and ran the tuned offline RL experiments. Joey ran the consistent offline experiments and online experiments. Both authors contributed equally to paper writing.

We thank John Schulman and Bo Dai for helpful discussions. Our research was supported by NSF(1651565), AFOSR (FA95501910024), ARO (W911NF-21-1-0125), ONR, CZ Biohub, and a Sloan Fellowship. Joey was supported by the Department of Defense (DoD) through the National Defense Science & Engineering Graduate (NDSEG) Fellowship Program.

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

# A  THE GUMBEL ERROR MODEL FOR MDPS

In this section, we functionally analyze Q-learning using our framework and further develop the Gumbel Error Model (GEM) for MDPs.

## A.1  RUST-MCFADDEN MODEL OF MDPS

For an MDP following the Bellman equations, we assume the observed rewards to be stochastic due to an unobserved component of the state. Let $\mathbf{s}$ be the observed state, and $(\mathbf{s}, \mathbf{z})$ be the actual state with hidden component $\mathbf{z}$. Then,

$$Q(\mathbf{s}, \mathbf{z}, \mathbf{a}) = R(\mathbf{s}, \mathbf{z}, \mathbf{a}) + \gamma \mathbb{E}_{\mathbf{s}' \sim P(\cdot|\mathbf{s}, \mathbf{a})}[\mathbb{E}_{\mathbf{z}'|\mathbf{s}'}[V(\mathbf{s}', \mathbf{z}')], \tag{15}$$

$$V(\mathbf{s}, \mathbf{z}) = \max_{\mathbf{a}} Q(\mathbf{s}, \mathbf{z}, \mathbf{a}). \tag{16}$$

**Lemma A.1.** *Given, 1) conditional independence (CI) assumption that $\mathbf{z}'$ depends only on $\mathbf{s}'$, i.e. $p(\mathbf{s}', z'|\mathbf{s}, \mathbf{z}, \mathbf{a}) = p(\mathbf{z}'|\mathbf{s}')p(\mathbf{s}'|\mathbf{s}, \mathbf{a})$ and 2) additive separablity (AS) assumption on the hidden noise: $R(\mathbf{s}, \mathbf{a}, \mathbf{z}) = r(\mathbf{s}, \mathbf{a}) + \epsilon(\mathbf{z}, \mathbf{a})$.*

*Then for i.i.d. $\epsilon(\mathbf{z}, \mathbf{a}) \sim \mathcal{G}(0, \beta)$, we recover the soft-Bellman equations for $Q(\mathbf{s}, \mathbf{z}, \mathbf{a}) = q(\mathbf{s}, \mathbf{a}) + \epsilon(\mathbf{z}, \mathbf{a})$ and $v(\mathbf{s}) = \mathbb{E}_z[V(\mathbf{s}, \mathbf{z})]$, with rewards $r(\mathbf{s}, \mathbf{a})$ and entropy regularization $\beta$.*

*Hence, a soft-MDP in MaxEntRL is equivalent to an MDP with an extra hidden variable in the state that introduces i.i.d. Gumbel noise in the rewards and follows the AS+CI conditions.*

*Proof.* We have,

$$q(\mathbf{s}, \mathbf{a}) = r(\mathbf{s}, \mathbf{a}) + \gamma \mathbb{E}_{\mathbf{s}' \sim P(\cdot|\mathbf{s}, \mathbf{a})}[\mathbb{E}_{\mathbf{z}'|\mathbf{s}'}[V(\mathbf{s}', \mathbf{z}')] \tag{17}$$

$$v(\mathbf{s}) = \mathbb{E}_{\mathbf{z}}[V(\mathbf{s}, \mathbf{z})] = \mathbb{E}_z[\max_{\mathbf{a}}(q(\mathbf{s}, \mathbf{a}) + \epsilon(\mathbf{z}))]. \tag{18}$$

From this, we can get fixed-point equations for $q$ and $\pi$,

$$q(\mathbf{s}, \mathbf{a}) = r(\mathbf{s}, \mathbf{a}) + \gamma \mathbb{E}_{\mathbf{s}' \sim P(\cdot|\mathbf{s}, \mathbf{a})}[\mathbb{E}_{\mathbf{z}'|\mathbf{s}'}[\max_{\mathbf{a}'}(q(\mathbf{s}', \mathbf{a}') + \epsilon(\mathbf{z}', \mathbf{a}'))]], \tag{19}$$

$$\pi(\cdot|s) = \mathbb{E}_{\mathbf{z}}[\underset{\mathbf{a}}{\arg\max}(q(\mathbf{s}, \mathbf{a}) + \epsilon(\mathbf{z}, \mathbf{a}))] \in \Delta_{\mathcal{A}}, \tag{20}$$

where $\Delta_{\mathcal{A}}$ is the set of all policies.

Now, let $\epsilon(\mathbf{z}, \mathbf{a}) \sim \mathcal{G}(0, \beta)$ and assumed independent for each $(\mathbf{z}, \mathbf{a})$ (or equivalently $(\mathbf{s}, \mathbf{a})$ due to the CI condition). Then we can use the Gumbel-Max trick to recover the soft-Bellman equations for $q(\mathbf{s}, \mathbf{a})$ and $v(\mathbf{s})$ with rewards $r(\mathbf{s}, \mathbf{a})$:

$$q(\mathbf{s}, \mathbf{a}) = r(\mathbf{s}, \mathbf{a}) + \gamma \mathbb{E}_{\mathbf{s}' \sim P(\cdot|\mathbf{s}, \mathbf{a})}[\mathbb{L}_{\mathbf{a}'}^{\beta}[q(\mathbf{s}', \mathbf{a}')]], \tag{21}$$

$$\pi(\cdot|s) = \underset{\mathbf{a}}{\text{softmax}}(q(\mathbf{s}, \mathbf{a})). \tag{22}$$

Thus, we have that the soft-Bellman optimality equation and related optimal policy can arise either from the entropic regularization viewpoint or from the Gumbel error viewpoint for an MDP.

**Corollary A.1.1.** *Converse: An MDP following the Bellman optimality equation and having a policy that is* softmax *distributed, necessarily has any i.i.d. noise in the rewards due to hidden state variables be Gumbel distributed, given the AS+CI conditions hold.*

*Proof.* McFadden (McFadden, 1972) proved this converse in his seminal work on discrete choice theory, that for i.i.d. $\epsilon$ satisfiying Equation 19 with a choice policy $\pi \sim \text{softmax}$ has $\epsilon$ be Gumbel distributed. And we show a proof here similar to the original for MDPs.

Considering Equation 20, we want $\pi(a|s)$ to be softmax distributed. Let $\epsilon$ have an unknown CDF $F$ and we consider there to be $N$ possible actions. Then,

$$P(\underset{\mathbf{a}}{\arg\max}(q(\mathbf{s}, \mathbf{a}) + \epsilon(z, \mathbf{a})) = \mathbf{a}_i|\mathbf{s}, \mathbf{z}) = P(q(\mathbf{s}, \mathbf{a}_i) + \epsilon(\mathbf{z}, \mathbf{a}_i) \geq q(\mathbf{s}, \mathbf{a}_j) + \epsilon(z, \mathbf{a}_j) \,\forall i \neq j \,|\mathbf{s}, \mathbf{z})$$

$$= P(\epsilon(\mathbf{z}, \mathbf{a}_j) - \epsilon(\mathbf{z}, \mathbf{a}_i) \leq q(\mathbf{s}, \mathbf{a}_i) - q(\mathbf{s}, \mathbf{a}_j) \,\forall i \neq j \,|\mathbf{s}, \mathbf{z})$$

Simplifying the notation, we write $\epsilon(\mathbf{z}, \mathbf{a}_i) = \epsilon_i$ and $q(\mathbf{s}, \mathbf{a}_i) = q_i$. Then $\epsilon_1, ..., \epsilon_N$ has a joint CDF $G$:

$$G(\epsilon_1, ..., \epsilon_N) = \prod_{j=1}^{N} P(\epsilon_j \leq \epsilon_i + q_i - q_j) = \prod_{j=1}^{N} F(\epsilon_i + q_i - q_j)$$

and we can get the required probability $\pi(i)$ as:

$$\pi(i) = \int_{\varepsilon=-\infty}^{+\infty} \prod_{j=1, j \neq i}^{N} F(\varepsilon + q_i - q_j) dF(\varepsilon) \tag{23}$$

For $\pi = \text{softmax}(q)$, McFadden (McFadden, 1972) proved the uniqueness of $F$ to be the Gumbel CDF, assuming translation completeness property to hold for $F$. Later this uniqueness was shown to hold in general for any $N \geq 3$ (Luce, 1977). □

## A.2 GUMBEL ERROR MODEL (GEM) FOR MDPs

To develop our Gumbel Error Model (GEM) for MDPs under functional approximation as in Section 3.1, we follow our simplified scheme of $M$ independent estimators $\hat{Q}$, which results in the following equation over $\bar{Q} = \mathbb{E}[\hat{Q}]$:

$$\bar{Q}_{t+1}(\mathbf{s}, \mathbf{a}) = r(\mathbf{s}, \mathbf{a}) + \gamma \mathbb{E}_{\mathbf{s}'|\mathbf{s}, \mathbf{a}}[\mathbb{E}_{\epsilon_t}[\max_{\mathbf{a}'}(\bar{Q}_t(\mathbf{s}', \mathbf{a}') + \epsilon_t(\mathbf{s}', \mathbf{a}'))]]. \tag{24}$$

Here, the maximum of random variables will generally be greater than the true max, i.e. $\mathbb{E}_\epsilon[\max_{\mathbf{a}'}(\bar{Q}(\mathbf{s}', \mathbf{a}') + \epsilon(\mathbf{s}', \mathbf{a}'))] \geq \max_{\mathbf{a}'} \bar{Q}(\mathbf{s}', \mathbf{a}')$ (Thrun & Schwartz, 1999). As a result, even initially zero-mean error can cause Q updates to propagate consistent overestimation bias through the Bellman equation. This is a known issue with function approximation in RL (Fujimoto et al., 2018).

Now, we can use the Rust-McFadden model from before. To account for the stochasticity, we consider extra unobserved state variables $z$ in the MDP to be the model parameters $\theta$ used in the functional approximation. The errors from functional approximation $\epsilon_t$ can thus be considered as noise added in the reward. Here, $CI$ condition holds as $\epsilon$ is separate from the dynamics and becomes conditionally independent for each state-action pair and $AS$ condition is implied. Then for $\bar{Q}$ satisfying Equation 24, we can apply the McFadden-Rust model, which implies that for the policy to be soft-optimal i.e. a softmax over $\bar{Q}$, $\epsilon$ will be Gumbel distributed.

Conversely, for the i.i.d. $\epsilon \sim \mathcal{G}$, $\bar{Q}(\mathbf{s}, \mathbf{a})$ follows the soft-Bellman equations and $\pi(\mathbf{a}|\mathbf{s}) = \text{softmax}(Q(\mathbf{s}, \mathbf{a}))$.

This indicates an optimality condition on the MDP – for us to eventually attain the optimal $\text{softmax}$ policy in the presence of functional boostrapping (Equation 24), the errors should follow the Gumbel distribution.

### A.2.1 TIME EVOLUTION OF ERRORS IN MDPs UNDER DETERMINISTIC DYNAMICS

In this section, we characterize the time evolution of errors in an MDP using GEM. We assume deterministic dynamics to simplify our analysis.

We suppose that we know the distribution of Q-values at time $t$ and model the evolution of this distribution through the Bellman equations. Let $Z_t(\mathbf{s}, \mathbf{a})$ be a random variable sampled from the distribution of Q-values at time $t$, then the following Bellman equation holds:

$$Z_{t+1}(\mathbf{s}, \mathbf{a}) = r(\mathbf{s}, \mathbf{a}) + \gamma \max_{\mathbf{a}'} Z_t(\mathbf{s}', \mathbf{a}'). \tag{25}$$

Here, $Z_{t+1}(\mathbf{s}, \mathbf{a}) = \max_{\mathbf{a}'}[r(\mathbf{s}, \mathbf{a}) + \gamma Z_t(\mathbf{s}', \mathbf{a}')]$ is a maximal distribution and based on EVT should eventually converge to an extreme value distribution, which we can model as a Gumbel.

Concretely, let's assume that we fix $Z_t(\mathbf{s}, \mathbf{a}) \sim \mathcal{G}(Q_t(\mathbf{s}, \mathbf{a}), \beta)$ for some $Q_t(\mathbf{s}, \mathbf{a}) \in \mathbb{R}$ and $\beta > 0$. Furthermore, we assume that the Q-value distribution is jointly independent over different state-actions i.e. $Z(\mathbf{s}, \mathbf{a})$ is independent from $Z(\mathbf{s}', \mathbf{a}')$ for $\forall (\mathbf{s}, \mathbf{a}) \neq (\mathbf{s}', \mathbf{a}')$. Then $\max_{\mathbf{a}'} Z_t(\mathbf{s}', \mathbf{a}') \sim \mathcal{G}(V(\mathbf{s}'), \beta)$ with $V(\mathbf{s}) = \mathbb{L}_{\mathbf{a}}^\beta[Q(\mathbf{s}, \mathbf{a})]$ using the Gumbel-max trick.

Then substituting in Equation 25 and rescaling $Z_t$ with $\gamma$, we get:

$$Z_{t+1}(\mathbf{s}, \mathbf{a}) \sim \mathcal{G}\left(r(\mathbf{s}, \mathbf{a}) + \gamma \mathbb{L}^\beta_{\mathbf{a}'}[Q(\mathbf{s}', \mathbf{a}')], \gamma \beta\right). \tag{26}$$

So very interestingly the Q-distribution becomes a Gumbel process, where the location parameter $Q(\mathbf{s}, \mathbf{a})$ follows the optimal soft-Bellman equation. Similarly, the temperature scales as $\gamma\beta$ and the distribution becomes sharper after every timestep.

After a number of timesteps, we see that $Z(\mathbf{s}, \mathbf{a})$ eventually collapses to the Delta distibution over the unique contraction $Q^*(\mathbf{s}, \mathbf{a})$. Here, $\gamma$ controls the rate of decay of the Gumbel distribution into the collapsed Delta distribution. Thus we get the expected result in deterministic dynamics that the optimal $Q$-function will be deterministic and its distribution will be peaked.

So if a Gumbel error enters into the MDP through a functional error or some other source at a timestep $t$ in some state $s$, it will trigger off an wave that propagates the Gumbel error into its child states following Equation 26. Thus, this Gumbel error process will decay at a $\gamma$ rate every timestep and eventually settle down with Q-values reaching the the steady solution $Q^*$. The variance of this Gumbel process given as $\frac{\pi^2}{6}\beta^2$ will decay as $\gamma^2$, similarly the bias will decay as $\gamma$-contraction in the $\mathcal{L}^\infty$ norm.

Hence, GEM gives us an analytic characterization of error propogation in MDPs under deterministic dynamics.

Nevertheless under stochastic dynamics, characterization of errors using GEM becomes non-trivial as Gumbel is not mean-stable unlike the Gaussian distribution. We hypothesise that the errors will follow some mix of Gumbel-Gaussian distributions, and leave this characterization as a future open direction.

## B  GUMBEL REGRESSION

We characterize the concentration bounds for Gumbel Regression in this section. First, we bound the bias on applying $\mathbb{L}^\beta$ to inputs containing errors. Second, we bound the PAC learning error due to an empirical $\hat{\mathbb{L}}^\beta$ over finite $N$ samples.

### B.1  OVERESTIMATION BIAS

Let $\hat{Q}(\mathbf{s}, \mathbf{a})$ be a random variable representing a Q-value estimate for a state and action pair $(\mathbf{s}, \mathbf{a})$. We assume that it is an unbiased estimate of the true Q-value $Q(\mathbf{s}, \mathbf{a})$ with $\mathbb{E}[\hat{Q}(\mathbf{s}, \mathbf{a})] = Q(\mathbf{s}, \mathbf{a})$. Let $Q(\mathbf{s}, \mathbf{a}) \in [-Q_{max}, Q_{max}]$

Then, $V(\mathbf{s}) = \mathbb{L}^\beta_{a\sim\mu}Q(\mathbf{s}, \mathbf{a})$ is the true value function, and $\hat{V}(\mathbf{s}) = \mathbb{L}^\beta_{a\sim\mu}\hat{Q}(\mathbf{s}, \mathbf{a})$ is its estimate.

**Lemma B.1.** *We have* $V(\mathbf{s}) \le \mathbb{E}[\hat{V}(\mathbf{s})] \le \mathbb{E}_{a\sim\mu}[Q(\mathbf{s}, \mathbf{a})] + \beta\log\cosh(Q_{max}/\beta)$.

*Proof.* The lower bound $V(\mathbf{s}) \le \mathbb{E}[\hat{V}(\mathbf{s})]$ is easy to show using Jensen's Inequality as $log\_sum\_exp$ is a convex function.

For the upper bound, we can use a reverse Jensen's inequality (Simić, 2009) that for any convex mapping $f$ on the interval $[a, b]$ it holds that:

$$\sum_i p_i f(x_i) \le f\left(\sum_i p_i x_i\right) + f(a) + f(b) - f\left(\frac{a+b}{2}\right)$$

Setting $f = -\log(\cdot)$ and $x_i = e^{\hat{Q}(\mathbf{s}, \mathbf{a})/\beta}$, we get:

$$\mathbb{E}_{\mathbf{a}\sim\mu}[-\log(e^{\hat{Q}(\mathbf{s},\mathbf{a})/\beta})] \le -\log(\mathbb{E}_{\mathbf{a}\sim\mu}[e^{\hat{Q}(\mathbf{s},\mathbf{a})/\beta}]) - \log(e^{Q_{max}/\beta}) - \log(e^{-Q_{max}/\beta}) + \log\left(\frac{e^{Q_{max}/\beta} + e^{-Q_{max}/\beta}}{2}\right)$$

On simplifying,

$$\hat{V}(\mathbf{s}) = \beta\log(\mathbb{E}_{\mathbf{a}\sim\mu}e^{\hat{Q}(\mathbf{s},\mathbf{a})/\beta}) \le \mathbb{E}_{\mathbf{a}\sim\mu}[\hat{Q}(\mathbf{s}, \mathbf{a})] + \beta\log\cosh(Q_{max}/\beta)$$

Taking expectations on both sides, $\mathbb{E}[\hat{V}(\mathbf{s})] \leq \mathbb{E}_{\mathbf{a}\sim\mu}[Q(\mathbf{s},\mathbf{a})] + \beta \log\cosh(Q_{max}/\beta)$. This gives an estimate of how much the LogSumExp overestimates compared to taking the expectation over actions for random variables $\hat{Q}$. This bias monotonically decreases with $\beta$, with $\beta = 0$ having a max bias of $Q_{max}$ and for large $\beta$ decaying as $\frac{1}{2\beta}Q_{max}^2$.

$\square$

## B.2 PAC LEARNING BOUNDS FOR GUMBEL REGRESSION

**Lemma B.2.** $\exp(\hat{\mathbb{L}}^\beta(X)/\beta)$ over a finite $N$ samples is an unbiased estimator for the partition function $Z^\beta = \mathbb{E}\left[e^{X/\beta}\right]$ and with a probability at least $1 - \delta$ it holds that:

$$\exp(\hat{\mathbb{L}}^\beta(X)/\beta) \leq Z^\beta + \sinh(X_{max}/\beta)\sqrt{\frac{2\log(1/\delta)}{N}}.$$

Similarly, $\hat{\mathbb{L}}^\beta(X)$ over a finite $N$ samples is a consistent estimator of $\mathbb{L}^\beta(X)$ and with a probability at least $1 - \delta$ it holds that:

$$\hat{\mathbb{L}}^\beta(X) \leq \mathbb{L}^\beta(X) + \frac{\beta\sinh(X_{max}/\beta)}{Z^\beta}\sqrt{\frac{2\log(1/\delta)}{N}}.$$

*Proof.* To prove these concentration bounds, we consider random variables $e^{X_1/\beta}, ..., e^{X_n/\beta}$ with $\beta > 0$, such that $a_i \leq X_i \leq b_i$ almost surely, i.e. $e^{a_i/\beta} \leq e^{X_i/\beta} \leq e^{b_i/\beta}$.

We consider the sum $S_n = \sum_{i=1}^N e^{X_i/\beta}$ and use Hoeffding's inequality, so that for all $t > 0$:

$$P(S_n - \mathbb{E}S_n \geq t) \leq \exp\left(\frac{-2t^2}{\sum_{i=1}^n\left(e^{b_i/\beta} - e^{a_i/\beta}\right)^2}\right) \tag{27}$$

To simplify, we let $a_i = -X_{max}$ and $b_i = X_{max}$ for all $i$. We also rescale t as $t = Ns$, for $s > 0$. Then

$$P(S_n - \mathbb{E}S_n \geq Ns) \leq \exp\left(\frac{-Ns^2}{2\sinh^2(X_{max}/\beta)}\right) \tag{28}$$

We can notice that L.H.S. is same as $P(\exp(\hat{\mathbb{L}}^\beta(X)/\beta) - \exp(\mathbb{L}^\beta(X)/\beta) \geq s)$, which is the required probability we want. Letting the R.H.S. have a value $\delta$, we get

$$s = \sinh(X_{max}/\beta)\sqrt{\frac{2\log(1/\delta)}{N}}$$

Thus, with a probability $1 - \delta$, it holds that:

$$\exp(\hat{\mathbb{L}}^\beta(X)/\beta) \leq \exp(\mathbb{L}^\beta(X)/\beta) + \sinh(X_{max}/\beta)\sqrt{\frac{2\log(1/\delta)}{N}} \tag{29}$$

Thus, we get a concentration bound on $\exp(\hat{\mathbb{L}}^\beta(X)/\beta)$ which is an unbiased estimator of the partition function $Z^\beta = \exp(\mathbb{L}^\beta(X)/\beta)$. This bound becomes tighter with increasing $\beta$, and asymptotically behaves as $\frac{X_{max}}{\beta}\sqrt{\frac{2\log(1/\delta)}{N}}$.

Similarly, to prove the bound on the log-partition function $\hat{\mathbb{L}}^\beta(X)$, we can further take $\log(\cdot)$ on both sides and use the inequality $\log(1 + x) \leq x$, to get a direct concentration bound on $\hat{\mathbb{L}}^\beta(X)$,

$$\hat{\mathbb{L}}^\beta(X) \leq \mathbb{L}^\beta(X) + \beta\log\left(1 + \sinh(X_{max}/\beta)e^{-\mathbb{L}^\beta(X)/\beta}\sqrt{\frac{2\log(1/\delta)}{N}}\right) \tag{30}$$

$$= \mathbb{L}^\beta(X) + \beta\sinh(X_{max}/\beta)e^{-\mathbb{L}^\beta(X)/\beta}\sqrt{\frac{2\log(1/\delta)}{N}} \tag{31}$$

$$= \mathbb{L}^\beta(X) + \frac{\beta\sinh(X_{max}/\beta)}{Z^\beta}\sqrt{\frac{2\log(1/\delta)}{N}} \tag{32}$$

This bound also becomes tighter with increasing $\beta$, and asymptotically behaves as $\frac{X_{max}}{Z^\beta}\sqrt{\frac{2\log(1/\delta)}{N}}$.

$\square$

## C   EXTREME Q-LEARNING

In this section we provide additional theoretical details of our algorithm, $\mathcal{X}$-QL, and its connection to conservatism in CQL (Kumar et al., 2020).

### C.1   $\mathcal{X}$-QL

For the soft-Bellman equation given as:

$$Q(\mathbf{s}, \mathbf{a}) = r(\mathbf{s}, \mathbf{a}) + \gamma \mathbb{E}_{\mathbf{s}' \sim P(\cdot|\mathbf{s}, \mathbf{a})} V(\mathbf{s}), \tag{33}$$

$$V(\mathbf{s}) = \mathbb{L}^\beta_{\mu(\cdot|s)}(Q(\mathbf{s}, \mathbf{a})), \tag{34}$$

we have the fixed-point characterization, that can be found with a recurrence:

$$V(\mathbf{s}) = \mathbb{L}^\beta_{\mu(\cdot|s)}\left(r(\mathbf{s}, \mathbf{a}) + \gamma \mathbb{E}_{\mathbf{s}' \sim P(\cdot|\mathbf{s}, \mathbf{a})} V(\mathbf{s})\right). \tag{35}$$

In the main paper we discuss the case of $\mathcal{X}$-QL under stochastic dynamics which requires the estimation of $\mathcal{B}^*$. Under deterministic dynamic, however, this can be avoided as we do not need to account for an expectation over the next states. This simplifies the bellman equations. We develop two simple algorithms for this case without needing $\mathcal{B}^*$.

**Value Iteration.**    We can write the value-iteration objective as:

$$Q(\mathbf{s}, \mathbf{a}) \leftarrow r(\mathbf{s}, \mathbf{a}) + \gamma V_\theta(\mathbf{s}'), \tag{36}$$

$$\mathcal{J}(\theta) = \mathbb{E}_{s \sim \rho_\mu, a \sim \mu(\cdot|s)}\left[e^{(Q(\mathbf{s}, \mathbf{a}) - V_\theta(\mathbf{s}))/\beta} - (Q(\mathbf{s}, \mathbf{a}) - V_\theta(\mathbf{s}))/\beta - 1\right]. \tag{37}$$

Here, we learn a single model of the values $V_\theta(\mathbf{s})$ to directly solve Equation 35. For the current value estimate $V_\theta(\mathbf{s})$, we calculate targets $r(\mathbf{s}, \mathbf{a}) + \gamma V_\theta(\mathbf{s})$ and find a new estimate $V'_\theta(\mathbf{s})$ by fitting $\mathbb{L}^\beta_\mu$ with our objective $\mathcal{J}$. Using our Gumbel Regression framework, we can guarantee that as $\mathcal{J}$ finds a consistent estimate of the $\mathbb{L}^\beta_\mu$, and $V_\theta(\mathbf{s})$ will converge to the optimal $V(\mathbf{s})$ upto some sampling error.

**Q-Iteration.**    Alternatively, we can develop a Q-iteration objective solving the recurrence:

$$Q_{t+1}(\mathbf{s}, \mathbf{a}) = r(\mathbf{s}, \mathbf{a}) + \gamma \mathbb{L}^\beta_{\mathbf{a}' \sim \mu}[Q_t(\mathbf{s}', \mathbf{a}')] \tag{38}$$

$$= r(\mathbf{s}, \mathbf{a}) + \mathbb{L}^{\gamma\beta}_{\mathbf{a}' \sim \mu}[\gamma Q_t(\mathbf{s}', \mathbf{a}')] \tag{39}$$

$$= \mathbb{L}^{\gamma\beta}_{\mathbf{a}' \sim \mu}[r(\mathbf{s}, \mathbf{a}) + \gamma Q_t(\mathbf{s}', \mathbf{a}')]. \tag{40}$$

where we can rescale $\beta$ to $\gamma\beta$ to move $\mathbb{L}$ out.

This gives the objective:

$$Q^t(\mathbf{s}, \mathbf{a}) \leftarrow r(\mathbf{s}, \mathbf{a}) + \gamma Q_\theta(\mathbf{s}', \mathbf{a}'), \tag{41}$$

$$\mathcal{J}(Q_\theta) = \mathbb{E}_{\mu(\mathbf{s}, a, \mathbf{s}')}\left[e^{(Q^t(\mathbf{s}, \mathbf{a}) - Q_\theta(\mathbf{s}, \mathbf{a}))/\gamma\beta} - (Q^t(\mathbf{s}, \mathbf{a}) - Q_\theta(\mathbf{s}, \mathbf{a}))/\gamma\beta - 1\right]. \tag{42}$$

Thus, this gives a method to directly estimate $Q_\theta$ without learning values, and forms our $\mathcal{X}$-TD3 method in the main paper. Note, that $\beta$ is a hyperparameter, so we can use an alternative hyperparameter $\beta' = \gamma\beta$ to simplify the above.

We can formalize this as a Lemma in the deterministic case:

**Lemma C.1.** *Let*

$$\mathcal{J}(\mathcal{T}_\mu Q - Q') = \mathbb{E}_{\mathbf{s}, \mathbf{a}, \mathbf{s}', \mathbf{a}' \sim \mu}\left[e^{(\mathcal{T}_\mu Q(\mathbf{s}, \mathbf{a}) - Q'(\mathbf{s}, \mathbf{a}))/\gamma\beta} - (\mathcal{T}_\mu Q(\mathbf{s}, \mathbf{a}) - Q'(\mathbf{s}, \mathbf{a}))/\gamma\beta - 1\right].$$

where $\mathcal{T}_\mu$ is a linear operator that maps $Q$ from current $(\mathbf{s}, \mathbf{a})$ to the next $(\mathbf{s}', \mathbf{a}')$: $\mathcal{T}_\mu Q(\mathbf{s}, \mathbf{a}) := r(\mathbf{s}, \mathbf{a}) + \gamma Q(\mathbf{s}', \mathbf{a}')$

Then we have $\mathcal{B}^* Q^t = \underset{Q' \in \Omega}{\operatorname{argmin}} \mathcal{J}(\mathcal{T}_\mu Q^t - Q')$, where $\Omega$ is the space of Q-functions.

*Proof.* We use that in deterministic dynamics,

$$\mathbb{L}_{\mathbf{a}' \sim \mu}^{\gamma\beta}[\mathcal{T}_\mu Q(\mathbf{s}, \mathbf{a})] = r(\mathbf{s}, \mathbf{a}) + \gamma \mathbb{L}_{\mathbf{a}' \sim \mu}^{\beta}[Q(\mathbf{s}', \mathbf{a}')] = \mathcal{B}^* Q(\mathbf{s}, \mathbf{a})$$

Then solving for the unique minima for $\mathcal{J}$ establishes the above results.

Thus, optimizing $\mathcal{J}$ with a fixed-point is equivalent to Q-iteration with the Bellman operator.

$\square$

## C.2 Bridging soft and conservative Q-Learning

**Inherent Convervatism in $\mathcal{X}$-QL** Our method is inherently conservative similar to CQL (Kumar et al., 2020) in that it underestimates the value function (in vanilla Q-learning) $V^\pi(\mathbf{s})$ by $-\beta \ \mathbb{E}_{\mathbf{a} \sim \pi(\mathbf{a}|\mathbf{s})}\left[\log \frac{\pi(\mathbf{a}|\mathbf{s})}{\pi_\mathcal{D}(\mathbf{a}|\mathbf{s})}\right]$, whereas CQL understimates values by a factor $-\beta \ \mathbb{E}_{\mathbf{a} \sim \pi(\mathbf{a}|\mathbf{s})}\left[\frac{\pi(\mathbf{a}|\mathbf{s})}{\pi_\mathcal{D}(\mathbf{a}|\mathbf{s})} - 1\right]$, where $\pi_\mathcal{D}$ is the behavior policy. Notice that the underestimation factor transforms $V^\pi$ in vanilla Q-learning into $V^\pi$ used in the soft-Q learning formulation. Thus, we observe that KL-regularized Q-learning is inherently conservative, and this conservatism is built into our method.

Furthermore, it can be noted that CQL conservatism can be derived as adding a $\chi^2$ regularization to an MDP and although not shown by the original work (Kumar et al., 2020) or any follow-ups to our awareness, the last term of Eq. 14 in CQL's Appendix B (Kumar et al., 2020), is simply $\chi^2(\pi\|\pi_\mathcal{D})$ and what the original work refers to as $D_{CQL}$ is actually the $\chi^2$ divergence. Thus, it is possible to show that all the results for CQL hold for our method by simply replacing $D_{CQL}$ with $D_{KL}$ i.e. the $\chi^2$ divergence with the KL divergence everywhere.

We show a simple proof below that $D_{CQL}$ is the $\chi^2$ divergence:

$$
\begin{aligned}
D_{CQL}\left(\pi, \pi_\mathcal{D}\right)(\mathbf{s}) &:= \sum_{\mathbf{a}} \pi(\mathbf{a} \mid \mathbf{s})\left[\frac{\pi(\mathbf{a} \mid \mathbf{s})}{\pi_\mathcal{D}(\mathbf{a} \mid \mathbf{s})} - 1\right] \\
&= \sum_{\mathbf{a}} (\pi(\mathbf{a} \mid \mathbf{s}) - \pi_\mathcal{D}(\mathbf{a} \mid \mathbf{s}) + \pi_\mathcal{D}(\mathbf{a} \mid \mathbf{s}))\left[\frac{\pi(\mathbf{a} \mid \mathbf{s})}{\pi_\mathcal{D}(\mathbf{a} \mid \mathbf{s})} - 1\right] \\
&= \sum_{\mathbf{a}} (\pi(\mathbf{a} \mid \mathbf{s}) - \pi_\mathcal{D}(\mathbf{a} \mid \mathbf{s}))\left[\frac{\pi(\mathbf{a} \mid \mathbf{s}) - \pi_\mathcal{D}(\mathbf{a} \mid \mathbf{s})}{\pi_\mathcal{D}(\mathbf{a} \mid \mathbf{s})}\right] + \sum_{\mathbf{a}} \pi_\mathcal{D}(\mathbf{a} \mid \mathbf{s})\left[\frac{\pi(\mathbf{a} \mid \mathbf{s})}{\pi_\mathcal{D}(\mathbf{a} \mid \mathbf{s})} - 1\right] \\
&= \sum_{\mathbf{a}} \pi_\mathcal{D}(\mathbf{a} \mid \mathbf{s})\left[\frac{\pi(\mathbf{a} \mid \mathbf{s})}{\pi_\mathcal{D}(\mathbf{a} \mid \mathbf{s})} - 1\right]^2 + 0 \text{ since, } \sum_{\mathbf{a}} \pi(\mathbf{a} \mid \mathbf{s}) = \sum_{\mathbf{a}} \pi_\mathcal{D}(\mathbf{a} \mid \mathbf{s}) = 1 \\
&= \chi^2(\pi(\cdot \mid \mathbf{s}) \| \pi_\mathcal{D}(\cdot \mid \mathbf{s})), \text{ using the definition of chi-square divergence}
\end{aligned}
$$

**Why $\mathcal{X}$–QL is better than CQL for offline RL** In light of the above results, we know that CQL adds a $\chi^2$ regularization to the policy $\pi$ with respect to the behavior policy $\pi_\mathcal{D}$, whereas our method does the same using the reverse-KL divergence.

Now, the reverse-KL divergence has a mode-seeking behavior, and thus our method will find a policy that better fits the mode of the behavior policy and is more robust to random actions in the offline dataset. CQL does not have such a property and can be easily affected by noisy actions in the dataset.

**Connection to Dual KL representation** For given distributions $\mu$ and $\pi$, we can write their KL-divergence using the dual representation proposed by IQ-Learn (Garg et al., 2021):

$$D_{KL}(\pi \| \mu) = \max_{x \in \mathbb{R}} \mathbb{E}_\mu[-e^{-x}] - \mathbb{E}_\pi[x] - 1,$$

which is maximized for $x = -\log(\pi/\mu)$.

We can make a clever substitution to exploit the above relationship. Let $x = (Q - \mathcal{T}^\pi \hat{Q}^k)/\beta$ for a variable $Q \in \mathbb{R}$ and a fixed constant $\mathcal{T}^\pi \hat{Q}^k$, then on variable substitution we get the equation:

$$\mathbb{E}_{s \sim \rho_\mu}[D_{KL}(\pi(\cdot|\mathbf{s}) \parallel \mu(\cdot|\mathbf{s}))] = \min_Q \mathcal{L}(Q), \text{ with}$$

$$\mathcal{L}(Q) = \mathbb{E}_{\mathbf{s} \sim \rho_\mu, \mathbf{a} \sim \mu(\cdot|\mathbf{s})} \left[ e^{(\mathcal{T}^\pi \hat{Q}^k(\mathbf{s},\mathbf{a}) - Q(\mathbf{s},\mathbf{a}))/\beta} \right] - \mathbb{E}_{\mathbf{s} \sim \rho_\mu, \mathbf{a} \sim \pi(\cdot|\mathbf{s})}[(\mathcal{T}^\pi \hat{Q}^k(\mathbf{s}, \mathbf{a}) - Q(\mathbf{s}, \mathbf{a}))/\beta] - 1$$

This gives us Equation 8 in Section 3.3 of the main paper, and is minimized for $Q = \mathcal{T}^\pi \hat{Q}^k - \beta \log(\pi/\mu)$ as we desire. Thus, this lets us transform the regular Bellman update into the soft-Bellman update.

## D EXPERIMENTS

In this section we provide additional results and more details on all experimental procedures.

### D.1 A TOY EXAMPLE

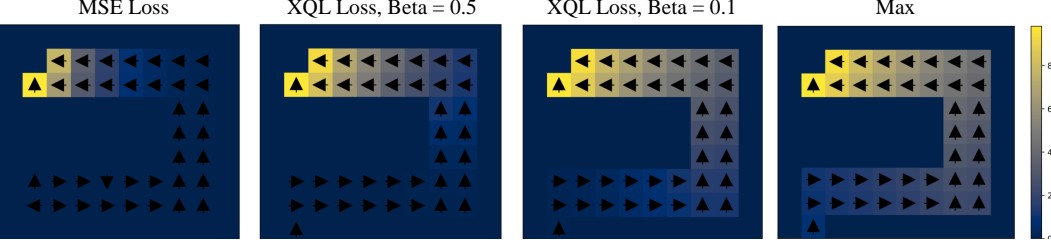

Figure 4: Here we show the effect of using different ways of fitting the value function on a toy grid world, where the agents goal is to navigate from the beginning of the maze on the bottom left to the end of the maze on the top left. The color of each square shows the learned value. As the environment is discrete, we can investigate how well Gumbel Regression fits the maximum of the Q-values. As seen, when MSE loss is used instead of Gumbel regression, the resulting policy is poor at the beginning and the learned values fail to propagate. As we increase the value of beta, we see that the learned values begin to better approximate the optimal max Q policy shown on the very right.

### D.2 BELLMAN ERROR PLOTS

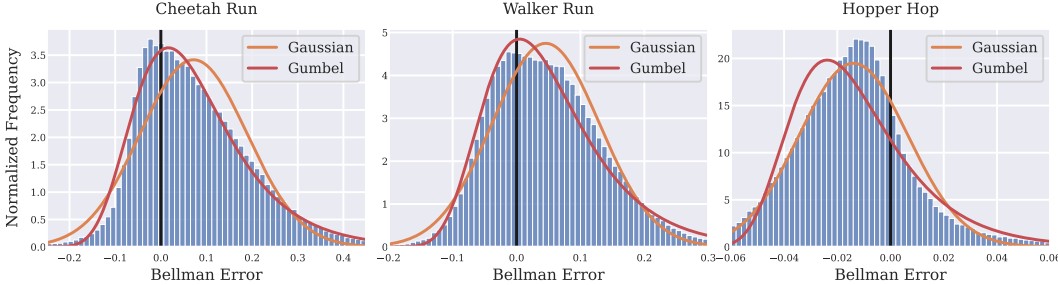

Figure 5: Additional plots of the error distributions of SAC for different environments. We find that the Gumbel distribution strongly fit the errors in first two environments, Cheetah and Walker, but provides a worse fit in the Hopper environment. Nonetheless, we see performance improvements in Hopper using our approach.

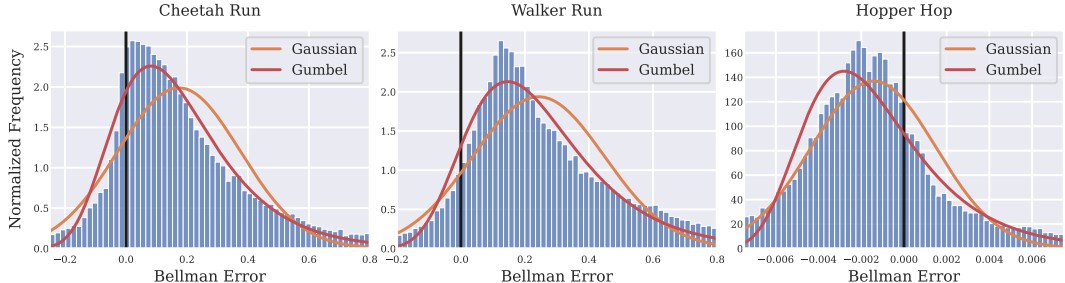

Figure 6: Plots of the error distributions of TD3 for different environments.

Additional plots of the error distributions for SAC and TD3 can be found in Figure 5 and Figure 6, respectively. Figure 1 and the aforementioned plots were generated by running RL algorithms for 100,000 timesteps and logging the bellman errors every 5,000 steps. In particular, the Bellman errors were computed as:

$$r(\mathbf{s}, \mathbf{a}) + \gamma Q_{\theta_1}(\mathbf{s}', \pi_\psi(\mathbf{s}')) - Q_{\theta_1}(\mathbf{s}, \mathbf{a})$$

In the above equation $Q_{\theta_1}$ represents the first of the two Q networks used in the Double Q trick. We do not use target networks to compute the bellman error, and instead compute the fully online quantity. $\pi_\psi(\mathbf{s}')$ represents the mean or deterministic output of the current policy distribution. We used an implementation of SAC based on Yarats & Kostrikov (2020) and an implementation of TD3 based on Fujimoto et al. (2018). For SAC we did the entropy term was not added when computing the error as we seek to characterize the standard bellman error and not the soft-bellman error. Before generating plots the errors were clipped to the ranges shown. This tended prevented over-fitting to large outliers. The Gumbel and Gaussian curves we fit using MLE via Scipy.

## D.3  NUMERIC STABILITY

In practice, a naive implementation of the Gumbel loss function $\mathcal{J}$ from Equation 11 suffers from stability issues due to the exponential term. We found that stabilizing the loss objective was essential for training. Practically, we follow the common max-normalization trick used in softmax computation. This amounts to factoring out $e^{\max_z z}$ from the loss and consequently scaling the gradients. This adds a per-batch adaptive normalization to the learning rate. We additionally clip loss inputs that are too large to prevent outliers. An example code snippet in Pytorch is included below:

```python
def gumbel_loss(pred, label, beta, clip):
    z = (label - pred)/beta
    z = torch.clamp(z, -clip, clip)
    max_z = torch.max(z)
    max_z = torch.where(max_z < -1.0, torch.tensor(-1.0), max_z)
    max_z = max_z.detach() # Detach the gradients
    loss = torch.exp(z - max_z) - z*torch.exp(-max_z) - torch.exp(-max_z)
    return loss.mean()
```

In some experiments we additionally clip the value of the gradients for stability.

## D.4  OFFLINE EXPERIMENTS

In this subsection, we provide additional results in the offline setting and hyper-parameter and implementation details.

Table 3 shows results for the Androit benchmark in D4RL. Again, we see strong results for $\mathcal{X}$-QL, where $\mathcal{X}$-QL-C with the same hyperparameters as used in the Franka Kitchen environments surpasses prior works on five of the eight tasks. Figure 7 shows learning curves which include baseline methods. We see that $\mathcal{X}$-QL exhibits extremely fast convergence, particularly when tuned. One issue however, is numerical stability. The untuned version of $\mathcal{X}$-QL exhibits divergence on the Antmaze environment.

We base our implementation of $\mathcal{X}$-QL off the official implementation of IQL from Kostrikov et al. (2021). We use the same network architecture and also apply the Double-$Q$ trick. We also apply the

Table 3: Evaluation on Adroit tasks from D4RL. $\mathcal{X}$-QL-C gives results with the same hyper-parameters used in the Franka Kitchen as IQL, and $\mathcal{X}$-QL-T gives results with per-environment $\beta$ and hyper-parameter tuning.

| Dataset | BC | BRAC-p | BEAR | Onestep RL | CQL | IQL | $\mathcal{X}$-QL C | $\mathcal{X}$-QL T |
|---|---|---|---|---|---|---|---|---|
| pen-human-v0 | 63.9 | 8.1 | -1.0 | - | 37.5 | 71.5 | **85.5** | **85.5** |
| hammer-human-v0 | 1.2 | 0.3 | 0.3 | - | 4.4 | 1.4 | 2.2 | **8.2** |
| door-human-v0 | 2 | -0.3 | -0.3 | - | 9.9 | 4.3 | **11.5** | **11.5** |
| relocate-human-v0 | 0.1 | -0.3 | -0.3 | - | **0.2** | 0.1 | 0.17 | 0.24 |
| pen-cloned-v0 | 37 | 1.6 | 26.5 | **60.0** | 39.2 | 37.3 | 38.6 | 53.9 |
| hammer-cloned-v0 | 0.6 | 0.3 | 0.3 | 2.1 | 2.1 | 2.1 | **4.3** | **4.3** |
| door-cloned-v0 | 0.0 | -0.1 | -0.1 | 0.4 | 0.4 | 1.6 | **5.9** | **5.9** |
| relocate-cloned-v0 | -0.3 | -0.3 | -0.3 | -0.1 | -0.1 | -0.2 | -0.2 | **-0.2** |

Figure 7: **Offline RL Results**. We show the returns vs number of training iterations for the D4RL benchmark, averaged over 6 seeds. For a fair comparison, we use batch size of 1024 for each method. XQL Tuned tunes the temperature for each environment, whereas XQL consistent uses a default temperature.

same data preprocessing which is described in their appendix. We additionally take their baseline results and use them in Table 1, Table 2, and Table 3 for accurate comparison.

We keep our general algorithm hyper-parameters and evaluation procedure the same but tune $\beta$ and the gradient clipping value for each environment. Tuning values of $\beta$ was done via hyper-parameter sweeps over a fixed set of values $[0.6, 0.8, 1, 2, 5]$ for offline save for a few environments where larger values were clearly better. Increasing the batch size tended to also help with stability, since our rescaled loss does a per-batch normalization. AWAC parameters were left identical to those in IQL. For MuJoCo locomotion tasks we average mean returns over 10 evaluation trajectories and 6 random seeds. For the AntMaze tasks, we average over 1000 evaluation trajectories. We don't see stability issues in the mujoco locomotion environments, but found that offline runs for the AntMaze environments could occasionally exhibit divergence in training for a small $\beta < 1$. In order to help mitigate this, we found adding Layer Normalization (Ba et al., 2016) to the Value networks to work well. Full hyper-parameters we used for experiments are given in Table 4.

### D.5 OFFLINE ABLATIONS

In this section we show hyper-parameter ablations for the offline experiments. In particular, we ablate the temperature parameter, $\beta$, and the batch size. The temperature $\beta$ controls the strength of KL penalization between the learned policy and the dataset behavior policy, and a small $\beta$ is beneficial for datasets with lots of random noisy actions, whereas a high $\beta$ favors more expert-like datasets.

Because our implementation of the Gumbel regression loss normalizes gradients at the batch level, larger batches tended to be more stable and in some environments lead to higher final performance. To show that our tuned $\mathcal{X}$-QL method is not simply better than IQL due to bigger batch sizes, we show a comparison with a fixed batch size of 1024 in Fig. 7.

### D.6 ONLINE EXPERIMENTS

We base our implementation of SAC off pytorch_sac (Yarats & Kostrikov, 2020) but modify it to use a Value function as described in Haarnoja et al. (2017). Empirically we see similar performance with and without using the value function, but leave it in for fair comparison against our $\mathcal{X}$-SAC variant. We base our implementation of TD3 on the original author's code from Fujimoto et al. (2018). Like in offline experiments, hyper-parameters were left as default except for $\beta$, which we tuned for each environment. For online experiments we swept over $[1, 2, 5]$ for $\mathcal{X}$–SAC and TD3. We found that these values did not work as well for TD3 - DQ, and swept over values $[3, 4, 10, 20]$. In online

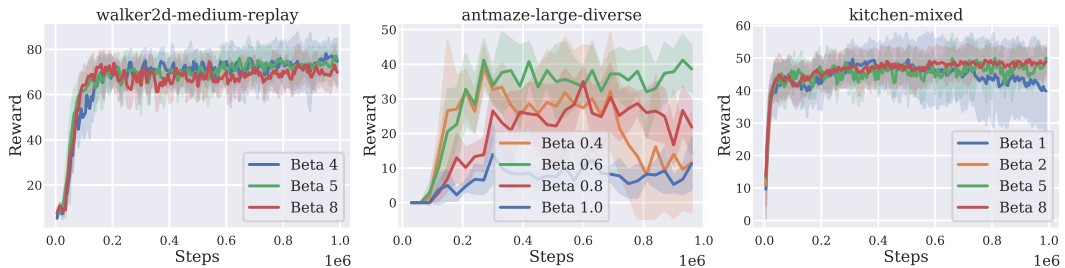

Figure 8: $\beta$ **Ablation**. Too large of a temperature $\beta$ and performance drops. When $\beta$ is too small, the loss becomes sensitive to noisy outliers, and training can diverge. Some environments are more sensitive to $\beta$ than others.

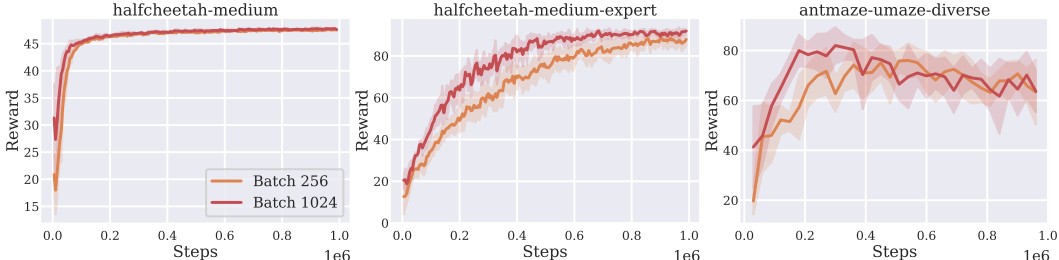

Figure 9: **Batch Size Ablation**. Larger batch sizes can make Gumbel regression more stable.

experiments we used an exponential clip value of 8. For SAC we ran three seeds in each environment as it tended to be more stable. For TD3 we ran four. Occasionally, our $\mathcal{X}$- variants would experience instability due to outliers in collected online policy rollouts causing exploding loss terms. We see this primarily in the Hopper and Quadruped environments, and rarely for Cheetah or Walker. For Hopper and Quadruped, we found that approximately one in six runs became unstable after about 100k gradient steps. This sort of instability is also common in other online RL algorithms like PPO due to noisy online policy collection. We restarted runs that become unstable during training. We verified our SAC results by comparing to Yarats & Kostrikov (2020) and our TD3 results by comparing to Li (2021) . We found that our TD3 implementation performed marginally better overall.

Table 4: Offline RL Hyperparameters used for $\mathcal{X}$–QL. The first values given are for the non per-environment tuned version of $\mathcal{X}$–QL, and the values in parenthesis are for the tuned offline results, $\mathcal{X}$–QL-T. V-updates gives the number of value updates per Q update, and increasing it reduces the variance of value updates using Gumbel loss on some hard environments.

| Env | Beta | Grad Clip | Batch Size | v_updates |
|---|---|---|---|---|
| halfcheetah-medium-v2 | 2 (1) | 7 (7) | 256 (256) | 1 (1) |
| hopper-medium-v2 | 2 (5) | 7 (7) | 256 (256) | 1 (1) |
| walker2d-medium-v2 | 2 (10) | 7 (7) | 256 (256) | 1 (1) |
| halfcheetah-medium-replay-v2 | 2 (1) | 7 (5) | 256 (256) | 1 (1) |
| hopper-medium-replay-v2 | 2 (2) | 7 (7) | 256 (256) | 1 (1) |
| walker2d-medium-replay-v2 | 2 (5) | 7 (7) | 256 (256) | 1 (1) |
| halfcheetah-medium-expert-v2 | 2 (1) | 7 (5) | 256 (1024) | 1 (1) |
| hopper-medium-expert-v2 | 2 (2) | 7 (7) | 256 (1024) | 1 (1) |
| walker2d-medium-expert-v2 | 2 (2) | 7 (5) | 256 (1024) | 1 (1) |
| antmaze-umaze-v0 | 0.6 (1) | 7 (7) | 256 (256) | 1 (1) |
| antmaze-umaze-diverse-v0 | 0.6 (5) | 7 (7) | 256 (256) | 1 (1) |
| antmaze-medium-play-v0 | 0.6 (0.8) | 7 (7) | 256 (1024) | 1 (2) |
| antmaze-medium-diverse-v0 | 0.6 (0.6) | 7 (7) | 256 (256) | 1 (4) |
| antmaze-large-play-v0 | 0.6 (0.6) | 7 (5) | 256 (1024) | 1 (1) |
| antmaze-large-diverse-v0 | 0.6 (0.6) | 7 (5) | 256 (1024) | 1 (1) |
| kitchen-complete-v0 | 5 (2) | 7 (7) | 256 (1024) | 1 (1) |
| kitchen-partial-v0 | 5 (5) | 7 (7) | 256 (1024) | 1 (1) |
| kitchen-mixed-v0 | 5 (8) | 7 (7) | 256 (1024) | 1 (1) |
| pen-human-v0 | 5 (5) | 7 (7) | 256 (256) | 1 (1) |
| hammer-human-v0 | 5 (0.5) | 7 (3) | 256 (1024) | 1 (4) |
| door-human-v0 | 5 (1) | 7 (5) | 256 (256) | 1 (1) |
| relocate-human-v0 | 5 (0.8) | 7 (5) | 256 (1024) | 1 (2) |
| pen-cloned-v0 | 5 (0.8) | 7 (5) | 256 (1024) | 1 (2) |
| hammer-cloned-v0 | 5 (5) | 7 (7) | 256 (256) | 1 (1) |
| door-human-v0 | 5 (5) | 7 (7) | 256 (256) | 1 (1) |
| relocate-human-v0 | 5 (5) | 7 (7) | 256 (256) | 1 (1) |

Table 5: Hyperparameters for online RL Algorithms

| Parameter | SAC | TD3 |
|---|---|---|
| Batch Size | 1024 | 256 |
| Learning Rate | 0.0001 | 0.001 |
| Critic Freq | 1 | 1 |
| Actor Freq | 1 | 2 |
| Actor and Critic Arch | 1024, 1024 | 256, 256 |
| Buffer Size | 1,000,000 | 1,000,000 |
| Actor Noise | Auto-tuned | 0.1, 0.05 (Hopper) |
| Target Noise | – | 0.2 |

Table 6: Values of temperature $\beta$ used for online experiments

| Env | $\mathcal{X}$-SAC | $\mathcal{X}$-TD3 | $\mathcal{X}$-TD3 - DQ |
|---|---|---|---|
| Cheetah Run | 2 | 5 | 4 |
| Walker Run | 1 | 2 | 4 |
| Hopper Hop | 2 | 2 | 3 |
| Quadruped Run | 5 | 5 | 20 |

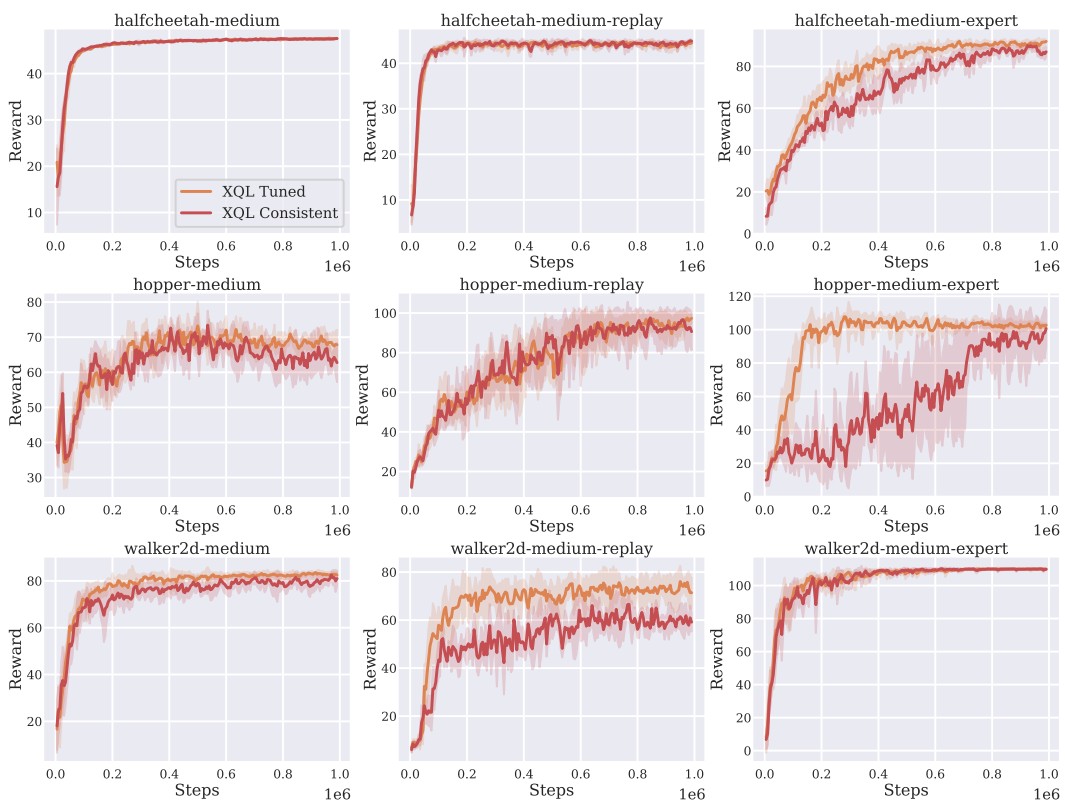

Figure 10: **Offline Mujoco Results**. We show the returns vs number of training iterations for the mujoco benchmarks in D4RL (Averaged over 6 seeds). $\mathcal{X}$-QL Tuned gives results after hyper-parameter tuning to reduce run variance for each environment, and $\mathcal{X}$-QL consistent uses the same hyper-parameters for every environment.

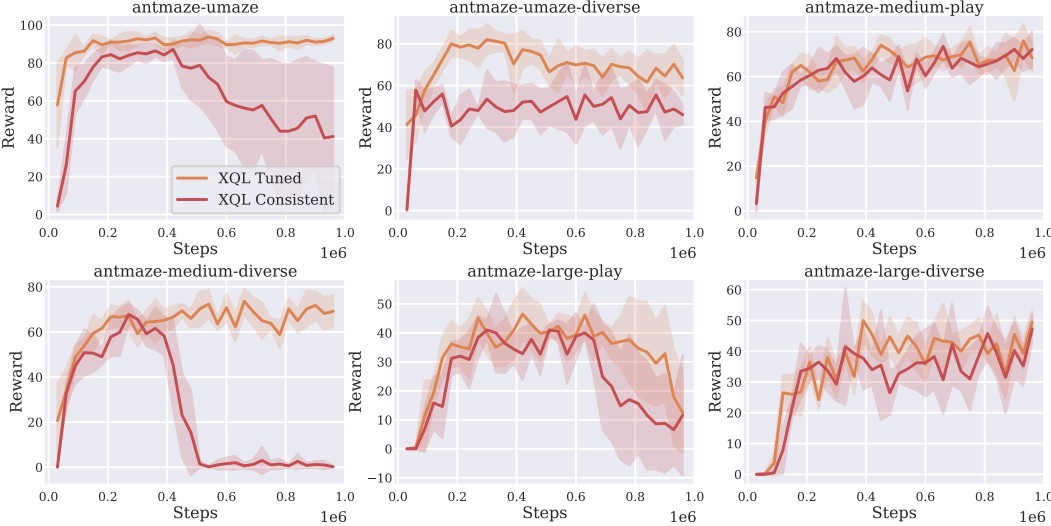

Figure 11: **Offline AntMaze Results**. We show the returns vs number of training iterations for the antmaze benchmarks in D4RL (Averaged over 6 seeds). $\mathcal{X}$-QL Tuned gives results after hyper-parameter tuning to reduce run variance for each environment, and $\mathcal{X}$-QL consistent uses the same hyper-parameters for every environment.

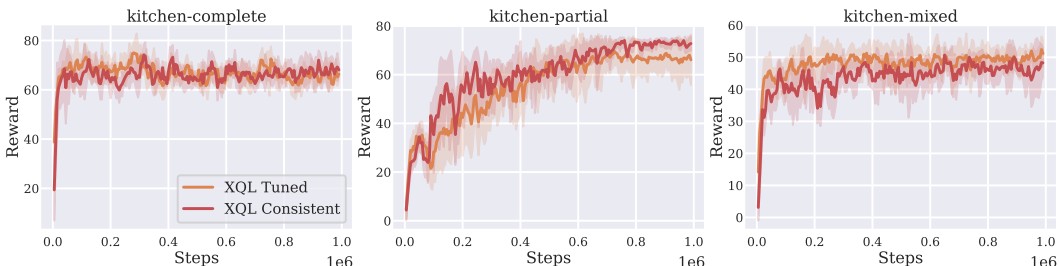

Figure 12: **Offline Franka Results**. We show the returns vs number of training iterations for the Franka Kitchen benchmarks in D4RL (Averaged over 6 seeds). $\mathcal{X}$-QL Tuned gives results after hyper-parameter tuning to reduce run variance for each environment, and $\mathcal{X}$-QL consistent uses the same hyper-parameters for every environment.

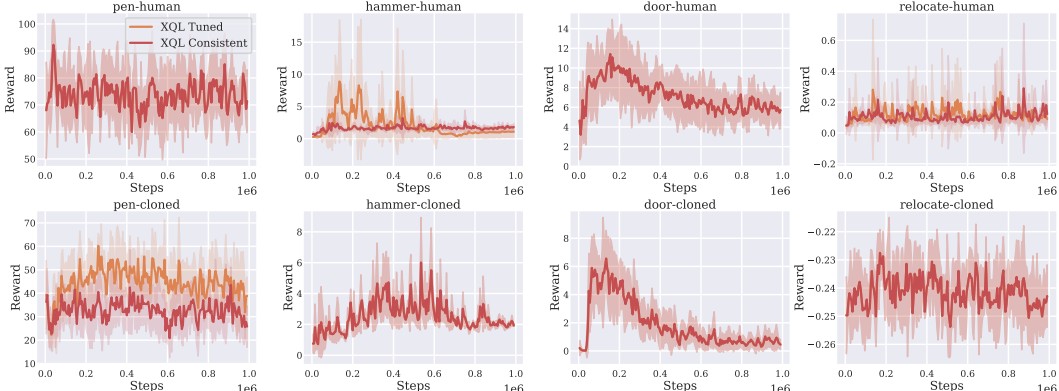

Figure 13: **Offline Andriot Results**. We show the returns vs number of training iterations for the Andriot benchmark in D4RL (Averaged over 6 seeds). $\mathcal{X}$-QL Tuned gives results after hyper-parameter tuning to reduce run variance for each environment, and $\mathcal{X}$-QL consistent uses the same hyper-parameters for every environment. On some environments the "consistent" hyperparameters did best.

