# OpenReview forum: "Extreme Q-Learning: MaxEnt RL without Entropy"
_ICLR.cc/2023/Conference — ICLR 2023 notable top 5%_

### Official Review · Reviewer_HuwQ · 2022-10-21

**Confidence:** 3
**Correctness:** 4
**Technical Novelty And Significance:** 3
**Empirical Novelty And Significance:** 3
**Recommendation:** 8

**Clarity, Quality, Novelty And Reproducibility:**

## Clarity
The paper is well-organized and easy to read.

## Quality
The technical part seems correct.

## Novelty
The paper makes an interesting observation and makes good practical contribution.

## Reproducibility
The submission includes a code base. The paper includes many implementation details.

**Strength And Weaknesses:**

# Strength
* The paper is very well organized, with sufficient background introduction so that readers without much previous knowledge could also understand the context.
* The usage of Gumbel regression is well organized, with good theoretical support.
* The paper also makes a good connection and shows how the minimizer of the Gumbel regression objective recovers previous update rules that require explicitly policy action distribution knowledge such as SAC or CQL, while XQL does not require the access to the policy during the value function update.
* The practical algorithm shows very promising improvement over previous methods, especially in the offline benchmarks. The practical algorithm seems also easy to be built on previous online methods.
# Weakness
* The motivation for using Gumbel regression seems not very obvious in the continuous action region because it's not obvious how to take the max operator. I am also confused about the presentation of the practical motivation (such as Fig.1) for two reasons:
1. The bellman error is recorded during the training. However, some value functions we learned during the training may not even represent a valid value function for any policy? Then what does this bellman error represent?
2. The bellman error is actually not calculated with max, but with the action that the corresponding policy takes. Again I understand this is due to the difficulty from the continuous action space, but this seems to deviate from the theoretical motivation.
* Although the paper claims that not requiring the access to the policy during value function update is a merit of the proposed algorithm, I could not see why this is significant in practice: in practice, we still need to train a policy anyway so we could assume we always have the access to the policy information?

**Summary Of The Paper:**

The paper makes an interesting observation that the error after the bellman optimality operation, instead of following gaussian noise, should in theory follow Gumbel distribution. This motivates the usage of Gumbel regression instead of least square to learn the Q functions, and the paper shows its connection to CQL in the policy evaluation regime. The paper also shows in practice one can introduce another V function and policy to deal with continuous action space, and the experiments show that the practical algorithm achieves promising results in both online and offline benchmarks.

**Summary Of The Review:**

Overall, this paper makes a very interesting observation and provides a new perspective on how we should perform Q learning. The method is well motivated, the paper is well-written, and the experiment results seem solid. Although there is still some remaining confusion, I still would like to recommend an acceptance for the paper.

---

> ### Author Response · Authors · 2022-11-11
> **Thank you for the review!**
>
> Thank you for your review of our work. We are happy to hear that the reviewer finds our work to be “promising” in several regards. We would like to address the concerns raised by the reviewer. Our approach is fundamentally designed to address the problem of fitting the max in continuous settings.
>
> *Motivation for Gumbel Regression in Continuous Settings* Gumbel regression is actually explicitly designed for the continuous setting specifically because “it’s not obvious how to take the max operator”. In particular, gumbel regression fits the Log-Sum-Exp of some distribution, and the log-sum-exp has tight bounds on the actual maximum depending on the temperature, or beta (https://en.wikipedia.org/wiki/LogSumExp) . By fitting a function approximator to the Log-Sum-Exp, we estimate the soft-maximum via the log-sum-exp, the same quantity used as the values in continuous MaxEnt RL.
>
> *Bellman Error Computation* We provide theoretical analysis of the bellman errors in Section 3.1. In order to see if this holds in practice, we look at the bellman errors observed during the course of training of standard RL algorithms. Like many things in deep learning, the theory does not perfectly reflect practice. To the comment on “not representing[ing] a valid value function for any policy”, this would be the case for all current RL algorithms. To the second comment on not computing the max, standard online RL algorithms (SAC, TD3) are proven to converge using policy improvement, and not by taking the max over actions. Because of this, the bellman error of SAC and TD3 is not the error from the optimal bellman backup. Because our approach actually fits the log-sum-exp, unlike prior methods it approximates the optimal bellman operator.
>
> *Accessing the Policy During Value Updates* Our approach can dis-entangle Q-learning and policy learning. While we do eventually want to obtain a policy, this loosens restrictions on the types of policies that can be learned. For example, prior works often use Gaussian policies because it is easy to compute the log-probability (and its gradient) of a Gaussian. We are not limited by this assumption, and future work can explore how this enables learning different types of policy distributions for which we do not have closed form probabilities.
>
> Please let us know if there are any more clarifying comments we can make!

---

### Official Review · Reviewer_pDUA · 2022-10-25

**Confidence:** 4
**Clarity, Quality, Novelty And Reproducibility:** Mentioned above
**Correctness:** 2
**Technical Novelty And Significance:** 3
**Empirical Novelty And Significance:** 2
**Recommendation:** 6

**Strength And Weaknesses:**

Strengths
- Incorporates the MaxEnt RL framework while avoiding the major problem of offline RL (extrapolation error from referring to OOD examples).
- Modeling TD-error as a Gumbel distribution seems to be more appropriate compared to modeling it as a Gaussian.

Weaknesses
- On offline RL, the proposed method still requires double-Q learning technique even though the method is expected to be more robust to Q-value overestimation.
- The experiments table for offline RL (Table 1) is misleading in several aspects:

First, the hyperparameters for the proposed method are tuned dataset-wise (refer to Table 4). However, previous works such as CQL or IQL keep their hyperparameters the same at least for each environment. For example, IQL fixes its hyperparameters the same for all MuJoCo locomotion tasks. Thus, the table results are not a fair comparison. Since the paper does not provide any hyperparameter sensitivity results, it is hard to conclude that the proposed method is actually the new "state-of-the-art" as the authors argue. Also, even if dataset-wise hyperparameter tuning is freely allowed, recent works seem to show higher performance on several datasets [1, 2].

Second, the runtime comparison also seems to be misleading. The authors note that the proposed method converges faster than IQL (Figure 6) and runs half the epochs compared IQL. However, based on the source code in the supplementary material, it seems that the proposed method uses a much larger batch size (1024) compared to IQL (256) [3]. Are the authors increasing the batch size and claiming the proposed method converges with much fewer iterations?

Questions
- On Lemma 3.4, why is the first expectation over \mu while the second expectation is over \pi?
- On Figure 3, what does -DQ on TD3 mean?

[1] An et al., Uncertainty-Based Offline Reinforcement Learning with Diversified Q-Ensemble, NeurIPS 2021.

[2] Cheng at al., Adversarially Trained Actor Critic for Offline Reinforcement Learning, ICML 2022.

[3] https://github.com/ikostrikov/implicit_q_learning

**Summary Of The Paper:**

This paper proposes a new Q-learning framework by formulating the TD-error as a Gumbel distribution rather than a Gaussian. This new formulation leads to MaxEnt-RL style RL algorithms, but without the need to sample from out-of-distribution examples. The proposed method shows fairly well performance on standard D4RL benchmarks.

**Summary Of The Review:**

This paper provides a new RL framework by modeling the TD-error as a Gumbel distribution, which allows avoiding sampling OOD datapoints while enjoying the advantages of MaxEnt RL. However, the experiments section seems to be misleading in several aspects.

---

> ### Author Response · Authors · 2022-11-11
> **We have overhauled Table 1 results, and more!**
>
> We would like to thank the reviewer for their in-depth review of our manuscript. In particular, we are grateful that the reviewer has pointed out ways of making the evaluation of XQL more fair. We are excited to address these points! To do so, we have changed the results in Table 1 to provide results with batch size 256 and per-domain tuning as well as our original per-environment tuned results. Below we hope to address this concern, and others highlighted in the review.
>
> *Overestimation* Because we are in the offline setting, we can still end up overestimating the Q values, especially since we are fitting a maximum. We found that clipped double Q learning helped increase stability in practice. This is consistent with IQL. Figure 3 of the ICLR version of IQL (https://openreview.net/pdf?id=68n2s9ZJWF8 ) shows that without clipped double Q learning, IQL becomes completely unstable. A central thesis of the EDAC paper is that larger ensembles for clipped Q learning are helpful for stability. While we believe the Gumbel loss function helps with overestimation, as supported by our results in comparison to IQL, it does not completely eliminate it. We do provide online experiments on X-TD3 without clipped double Q-learning in Section 4.2 (that is what the -DQ stands for).
>
> *Recent Work*: Thank you for bringing these recent works to our attention! ATAC, published at ICML in July, is indeed very exciting concurrent work.  We have changed the text to be more specific about which environments we achieve new state-of-the-art results, specifically the Franka Kitchen tasks. It is worth noting that we also see very consistent performance across environments. For example, we don’t have the highest absolute score in every mujoco environment when compared to base, but our results are very consistent, scoring within margin of the best baseline or surpassing it in 7/9 locomotion tasks using new domain tuned hyper-parameters (not dataset specific).
>
> EDAC is also exciting. To our understanding, the core of the method is based around learning an ensemble of Q values. We view this ensemble-based advancement as complementary to our approach as it could be integrated with XQL. Our current results compare against a large set of algorithms that vary in loss computation, our core contribution.
>
> *Overhaul of Table 1*: We have adjusted the results in Table 1 to show results with more consistent hyper-parameters to the same level of tuning as IQL. There are now two columns: XQL Tuned, and XQL. XQL Tuned are the previous results with more specific per-environment tuning. The XQL column has results using a batch size of 256 and the same beta value for each domain: 2 for all of mujoco, 0.6 for all of antmaze, and 5 for all of Franka Kitchen. Both methods use the same AWAC hyperparameters as IQL. The learning curves in the appendix have also been updated. Beta is a temperature parameter, as in SAC, that is dependent on the entropy of the actions in the dataset and directly controls the magnitude of our loss, making it an important parameter. It was tuned environment, or dataset-wise in the original SAC publication. We now provide both sets of results in Table 1.
>
> *Convergence Time*
> We have updated the convergence time in Table 1 and Figure 6 using the non-tuned version of XQL with batch-size 256. We found that using a batch size of 1024 was slightly more robust because our loss function uses a per-batch normalization for stability. With GPU acceleration, a larger batch size tended to not affect speed that much.
>
> *Ablations* We have added two additional ablations to the appendix, over both the batch size and temperature parameter beta.
>
> *Lemma 3.4*: That’s a typo! The expectations should be under $\mu$, thanks for pointing this out!
>
> *What is - DQ*: “- DQ” was meant to refer to a version of TD3 that does not use the double-Q trick aka minus Double Q. We believe this is an aspect of the experiments the reviewer would be interested in per their comments on Table 1.
>
> With these new results we believe we have improved the experimental evaluation of our method. Please let us know if there are any remaining questions or concerns.

---

> > ### Comment · Reviewer_pDUA · 2022-11-16
> > **Response to Authors**
> >
> > Thanks for the detailed response!
> >
> > I read the response and the reviews from other reviewers. The response addresses most of my concerns. I especially appreciate the authors' effort done on hyperparameter sensitivity experiments and addressing overstatements from the original script. I am raising my score.
> >
> > Minor comment: I recommend the authors to also update the SOTA-related statements on the intro (page 2, "Our resulting algorithms ~") and the conclusion to make them more consistent to the updated experiment section.

---

> ### Author Response · Authors · 2022-11-16
> **Looking for Feedback**
>
> Hi! Thank you again for your review! We would like to know if you have had a chance to look at the updates we have made to address the concerns raised in your review. There are only two or three days left in the discussion period, and as such we are running out of time to run any additional experiments.
>
> Best,
> Authors

---

### Official Review · Reviewer_PnpD · 2022-10-26

**Confidence:** 3
**Correctness:** 4
**Technical Novelty And Significance:** 3
**Empirical Novelty And Significance:** 3
**Recommendation:** 10

**Clarity, Quality, Novelty And Reproducibility:**

The paper is clear, quality is high, novelty is high. Reproducibility is clear based on equations and empirical explanations.



**Strength And Weaknesses:**

Strengths:
- The paper motivates their algorithm based on theoretical foundations and provides Lemmas to support their final loss functions.
- The paper is well-written and easy to follow.
- Empirical results against strong baselines show significant improvement

Weaknesses:
- As far as I can tell, hyper-parameter tuning is done on the test set, there is no separate validation set.
- It will be great to have this algorithm available as open source.

**Summary Of The Paper:**

The paper proposes Gumbel-regression as an alternative to mean squared error regression for value functions in reinforcement learning. The use of Gumbel distribution is motivated from established theory and empirical observations. The resulting algorithm outperforms state-of-the-art in both online and offline RL benchmarks.

**Summary Of The Review:**

No major weaknesses as such, enjoyed reading this paper. It will help to clarify why tuning Beta on each environment does not lead to over-fitting.

---

> ### Author Response · Authors · 2022-11-11
> **Thank you for the review!**
>
> We would like to thank the reviewer for finding our work to “show strong improvements” and be “easy to follow”. We would like to address the reviewers question about tuning Beta.
>
> It is true that tuning beta for each environment can improve performance. In order to more fairly compare with prior work we have changed the evaluation in Table 1 to be split into two parts: XQL-T (Tuned) gives the previously shown results where the value of Beta was tuned for each environment, and XQL, which is now a set of new results with consistent hyperparameters for each domain and the same batch size as prior work. This serves as a direct comparison between our gumbel loss and prior work. Even under more even hyperparameters, we still outperform prior works and consistently show higher performance across domains showing that we did not just overfit beta to each environment.
>
> We believe it is worth pointing out that tuning conservatism and temperature parameters remains an open challenge. Many offline algorithms (IQL, CQL, TD3+BC) require tuning a conservatism parameter, and online max entropy algorithms (SAC) also employ tuning strategies.

---

### Official Review · Reviewer_XF2J · 2022-11-02

**Confidence:** 3
**Clarity, Quality, Novelty And Reproducibility:** Paper was easy to read, and I do not …
**Correctness:** 3
**Technical Novelty And Significance:** 3
**Empirical Novelty And Significance:** Not applicable
**Recommendation:** 6

**Strength And Weaknesses:**

I think this paper brings forth some very interesting ideas to the table. Using Gumbel regression to model Q updates is to the best of my knowledge quite novel and in my opinion definitely something worth further exploration. The paper overall was quite easy to read and the authors did a very good job of justifying most claims and the technical choices involved in the algorithm. Some of my comments on some specific points in the paper:
- Intuitively I agree with the authors in that I think this EVT-based approach would work quite well in an offline setting (and experiment results do support this) since it naturally introduces conservatism into the algorithm. However I would definitely like to understand better where the source of improvement comes from and especially some additional comparisons with other algorithms in this regard, this could come in the form of a toy example that gives easier visualization.
- For the online setting, I'm not fully convinced why this approach would be better than an algorithm like SAC. Performance for online RL do not seem to offer much improvements. The environments used in the paper are generally considered to be fairly easy environments and the number of random seeds used is very small (3 or 4). Also unlike PPO, numerical instability isn't usually a huge issue for vanilla TD3/SAC for these particular environments.
- Adding to the above point, it does seem that numerical instability is quite a major issue from a practical perspective, though the authors have introduced ways to mitigate this, I feel that this could still be a huge obstacle to wide practical adoption of this approach.
- One thing you mentioned in the conclusion section about potential future directions is to integrate automatic parameter tuning for the temperature. Could you elaborate on some of the challenges for doing this and why using the mechanism for this introduced in [1] would be difficult?

[1] Haarnoja, Tuomas, et al. "Soft actor-critic algorithms and applications." arXiv preprint arXiv:1812.05905 (2018).



**Summary Of The Paper:**

This work introduces a new class of Q-learning-based algorithm for both online and offline RL for continuous control tasks based on extreme value theory. By using the insight that the maximum of i.i.d. random variables with exponential tails has a Gumbel distribution, the authors derived a new set of update rules which are equivalent to using soft Bellman backups but do not involve the use of entropies. The effectiveness of the algorithm is demonstrated in both online and offline settings using a set of standard benchmarks.

**Summary Of The Review:**

Overall, I think the community has a lot to gain from the ideas introduced in this paper, which is why I am recommending this paper for acceptance. However having said that, I do think the authors could have done a better job in justifying why "without entropy" is better and I hope the authors can offer some additional insight in the discussion period.

---

> ### Author Response · Authors · 2022-11-11
> **Thank you for your review!**
>
> Thanks for your time and effort in the assessment of our work. We are delighted that you found the work to be “quite novel” and “definitely something worth further exploration”. We would like to address any lingering questions about the draft.
>
> *A Toy Example*: We are working on getting together a toy illustration to include in the appendix, and will include it if we have time before the end of the rebuttal period. There are two main motivations for why our method should work well: 1) maximum entropy RL has shown to be very performant, and we extend it to the offline setting, and 2) similar to IQL we estimate a near optimal value function, which is likely more performant than one step style approaches (See IQL Kostrikov et al. 2021 Figure 2).
>
> *The Online Setting*: We agree that the inherent conservatism of our algorithm is slightly better suited for the offline setting. That being said, we show that we can approximate the type of trust region style updates commonly used in on-policy algorithms. As shown in Figure 3, this effect is modest in most environments except for in Hopper-Hop, where we achieve substantially better performance. This is likely because rewards in the Hopper environment are more sparse and brittle than in the other environments, and thus policy learning benefits from more trust-region regularization. We do not claim to show more than marginal performance gains in the online setting. That being said, X-QL solves the max-entropy problem with a completely different formulation than SAC and still gets the same or better performance. Again, this can be seen to be a strength and not a weakness. Modifying existing methods to use Gumbel regression is simple and theoretically grounded, making it an appealing choice for practitioners. Our approach also does not explicitly require access to the policy distribution for its soft-objective (Equation 11), unlike online RL works like SAC, which could lead to interesting follow-ups in the future.
>
> *Numerical Stability*: Though it would be amazing if our theory could translate directly into practice, the exponential nature of the Gumbel Loss function does make using it a bit more difficult. As one can imagine, taking the exponential of the Q-function can cause exploding gradient issues. We introduced a number of changes to combat this, including the method used for computing the loss outlined in the Appendix. Nevertheless, after our changes we only see this effect when training with very small betas. This instability was not noticeable in the mujoco or kitchen environments. In order to better understand the stability of our method, we have introduced ablations over the batch size and temperature parameters in the appendix. In many environments, our method is robust to the value of beta. However, consistent with prior works like IQL, the value of Beta needs to be chosen carefully to maximize performance in some environments. For example, the performance IQL completely collapses in AntMaze if the expectile is chosen to  be 0.7 instead of 0.9 (see https://openreview.net/pdf?id=68n2s9ZJWF8 Figure 3). Our method appears to be as or more stable than IQL in this regard as shown in the ablation over beta in Appendix D.
>
> *Automatic Parameter Tuning*: The automatic entropy tuning used in Haarnoja et al. 2018 requires computing the log probability of actions under the current policy. One appealing facet of our method is that it does not  require explicitly computing the log-probabilities of the policy, adding this type of parameter tuning would remove that benefit. Moreover, this approach uses a state-dependent policy variance. In practice, this is not used in offline RL algorithms (IQL, etc.) because of stability. It would be hard to describe our work as learning soft-optimal policies without doing explicit entropy calculation if we used the automatic tuning scheme from Haarnoja et al. 2018. Moreover, automatic conservatism tuning (equivalent to tuning beta) has never been done before by any offline RL methods. CQL, IQL, and TD3+BC both have a tuned hyperparameter that balances the level of conservatism. We believe determining how to do this effectively would warrant its own publication.

---

> ### Author Response · Authors · 2022-11-17
> **Added a toy example**
>
> Hi, we would just like to let you know that we have added a toy example to the first portion of Appendix D that demonstrations how our XQL method fits the soft-optimal values in comparison to MSE loss and Max Q. We hope this serves as a visualization of how our method helps approximate the maximum value in continuous settings.

---

### Public Comment · ~Sobhan_Mohammadpour1 · 2022-11-07
**Missing citations?**

https://www.atlantis-press.com/journals/jsta/125935600/view (near equation 5) showed that the LINEX function (eq. 8 in this paper) corresponds to a Gumbel regression, i think they should cite that paper and also call the function LINEX as it's fairly well known in the statistical community. Otherwise it was a very interesting read.

---

> ### Author Response · Authors · 2022-11-11
> **Thanks!**
>
> Thank you for pointing this out! We have added citations to this, and noted that our Gumbel regression is also known as Linex in the statistics community.

---

### Public Comment · ~Outongyi_Lv1 · 2023-04-07
**I have read this paper carefully, but I would like to ask the author some questions**

Hello, authors, I found some small mistakes in the article that can be corrected:

1、In Theorem 1:limn→∞ maxi(Xi) should be limi→∞ maxi(Xi)

2、In 3.1 GUMBEL ERROR MODEL section:the error formula is wrong: r(s, a) − γQ(s′, π(s′)) − Q(s, a) should be: r(s, a) + γQ(s′, π(s′)) − Q(s, a)

In addition I have some doubts for this paper. Can the authors help me? 1、 The authors propose a very novel approach called "Gumbel" regression, The author claims that ϵt(s, a) is subject to the Gumbel distribution when t limit to ∞, but in Q updating, The maximum value is taken for the action "a", I follow the author's derivation, but in my derivation： ϵ_{t+1}(s, a)=max_{a'}[Q^{true}(s',a')+ϵ_{t}(s, a)]-E[max_{a'}[Q^{true}(s',a')+ϵ_{t}(s, a)]=max_{a'}[M(s'a',ϵ_t)] and take limit, ϵ(s, a)=max_{a'}[M(s'a',ϵ)],but if ϵ(s, a)～Gumbel distribution, the action must be continuous. if the action is discrete, such as: up, down, left, and right four actions, why ϵ(s, a)～Gumbel distribution? How to proof this? Can you tell me How to get it directly by combining Theorem 1 with formula (5)? 2、 Can the author give in detail how (13) is obtained by minimizing the Reverse-KL divergence? I have deduced it for a long time, maybe he is consistent with the derivation of SAC? In addition, why does
π
 in (12) take log? In fact, why can't it be regarded as the expectation about \pi, and then take argmax, isn't the effect the same? Is it to unify with formula (13)?

I am very interested in this paper, and I hope to continue my research on this basis. I also think the idea of this paper is very innovative. I hope that the author can correct and answer the doubts and small mistakes in the paper. Maybe The question looks very "idiot", please forgive me, thanks!

---

> ### Author Response · Authors · 2023-04-13
> **Thank you for your comments!**
>
> Hi!
> Thank you for engaging with our work! I'll try to answer your questions as best I can!
>
> Small mistakes:
> 1. I believe theorem 1 is correct. Namely, assume we have a set of random variables $X_1, ... X_n$. We consider $\max (X_1, ... X_n)$ which we short-hand as $\max_i X_i$.
> 2. This is indeed a mistake! Thank you for pointing this out. It doesn't seem like we can update the draft at this point, but we will try to make the change in our next ArXiv update.
>
> For the questions about the Gumbel distribution, we assume continuous actions. Div (my co-author) might be better suited to answer questions about the Gumbel Error model in Section 3.1.
>
> Below I'll try to write out the full derivation of the policy learning objectives via the KL divergence. We use the fact that $\pi^*(a|s) \propto \mu(a|s) E^{Q^*(s,a) - V^*(s)}$ and assume that we want to learn $\pi$. Moreover, we use the convention that $D_{KL}(\pi^*||\pi)$ is the forward KL divergence as we are learning $\pi$ and $D_{KL}(\pi||\pi^*)$ to be the reverse divergence. We find that this is consistent with prior literature, where the learned distribution is considered second (AWAC: Accelerating Online RL with Offline Data. Nair Gupta et al, https://dibyaghosh.com/blog/probability/kldivergence.html). Below is a (non-rigorous) outline of the derivations. For simplicity I have omitted the $\beta$ term, but it can easily be added back in.
>
> Forward KL:
> $$\min_\pi D_{KL}(\pi^*||\pi) = \min_\pi \mathbb{E}_{\pi^*} \left[ \log  \frac{\mu(a|s)e^{Q^*(s,a) - V^*(s)}}{\pi(a|s)} \right] $$
>
> $$ = \min_\pi \mathbb{E}_{\pi^*} \left[- \log \pi(a|s)\right]$$
>
> $$ = \min_\pi \mathbb{E}_{\mu} \left[- e^{Q^*(s,a) - V^*(s)} \log \pi(a|s)\right]$$
>
> Reverse KL:
> $$\min_\pi D_{KL}(\pi||\pi^*) = \min_\pi \mathbb{E}_\pi \left[\log\frac{\pi(a|s)}{\mu(a|s)e^{Q^*(s,a) - V^*(s)}}\right]$$
> $$= \min_\pi \mathbb{E}_\pi \left[ \log\pi(a|s) - \log \mu(a|s) - Q^*(s,a) - V^*(s) \right] $$
> $$= \min_\pi \mathbb{E}_\pi \left[ \log\frac{\pi(a|s)}{\log \mu(a|s)} - Q^*(s,a) \right] $$
> $$= \max_\pi \mathbb{E}_\pi \left[Q^*(s,a) -  \log\frac{\pi(a|s)}{\mu(a|s)}  \right] $$
> In the above $a \sim \pi$, thus when we maximize $Q^*(s,a)$ we can compute it as $Q^*(s,\pi(s))$.
>
> I hope this helps!

---

> > ### Public Comment · ~Outongyi_Lv1 · 2023-04-14
> > **Thank you for your reply!**
> >
> > I am very grateful to the author for answering my doubts and questions.
> > The authors explained very clearly, thank you!

---

### Public Comment · ~Outongyi_Lv1 · 2023-04-08
**In addition, I have discovered a fatal error in the article, and I hope the author can correct it soon.**

In Lemma 3.4, this paper

L(Q) = Es∼ρµ,a∼µ(·|s) [ e(T π ˆQk(s,a)−Q(s,a))/βi ]− Es∼ρµ,a∼µ(·|s)
[(T π ˆQk(s, a) − Q(s, a))/β] − 1

should be:

L(Q) = Es∼ρµ,a∼µ(·|s) [ e(T π ˆQk(s,a)−Q(s,a))/βi ]− Es∼ρµ,a∼π(·|s)
[(T π ˆQk(s, a) − Q(s, a))/β] − 1

Otherwise, you won't get the update rule in this paper because this is not any π in expectations


In addition, I would like to ask the authors how to use greedy strategies? For example, in other Q learning methods such as DDPG, although it can be offline, I can still learn this network by adding some noise to the action, otherwise it is easy to jump into the local optimal solution, But in this paper, the authors point out that the offline algorithm is only based on the existence (s,a,s') from the original data? Won't this fall into the local optimal solution? I tried programming implementation and found that the effect is significantly different from offline DDPG algorithm. Can the author help me provide an explanation or the code?

---

> ### Author Response · Authors · 2023-04-26
> **Thanks for your questions!**
>
> Hi! I'm not sure if I completely follow your comments on Lemma 3.4. Lemma 3.4 is derived from taking the Gumbel Loss function in expectation under the reference distribution $\mu$., so I believe the Lemma is correct. How $\mu$ is selected changes the effect of the algorithm.
>
> In offline settings, $\mu$ is chosen to only be the offline data.
>
> In online settings, we choose $\mu$ to be the last policy, which in practice is just the policy before the gradient update. This corresponds to sampling actions from the policy when doing the ExtremeQ value update. In practice, we include entropy regularization when learning the policy. This means that in practice, we do include action noise when estimating the values. In the code you can see [here](https://github.com/Div99/XQL/blob/main/online/research/algs/gumbel_sac.py#L200) that we sample actions from the policy distribution (there is also code for adding extra noise on top of the policy's samples which we set to 0 in practice), and use that in our value function update.
>
> I hope this helps clarify your questions!

---

> > ### Public Comment · ~Outongyi_Lv1 · 2023-06-08
> > **Thank you for your answer!**
> >
> > Thank you very much for your answer！ I will analyze it more carefully,
> > By the way,The Gumbel distribution pdf function in paper seems to be missing a 1/beta?
> > Best wishes.

---

### Decision · Program_Chairs · 2023-01-20

**Decision:**

Accept: notable-top-5%

**Justification For Why Not Higher Score:**

n/a

**Justification For Why Not Lower Score:**

This seems to me like a simultaneously very innovative and very practical contribution.  It is clearly above bar for acceptance.

**Metareview: Summary, Strengths And Weaknesses:**

(a) Summary: This paper exploits extreme value theory to derive a new set of updates that are equivalent to a soft Bellman backup, but which do not require the explicit computation of entropy.

(b) Strengths: The proposed method has promising performance on both online and offline benchmarks. It is well-grounded theoretically, and uses the theoretical observation (errors should be modeled as Gumbel-distributed rather than Gaussian-distributed) to derive important practical consequences.

(c) Weaknesses: Some reviewers had some clarity concerns around the empirical evaluation that appear to have been resolved in rebuttal/revision; there were concerns about the need to set hyperparameters and how they were chosen in the evaluation.  The applicability to the online setting seemed less clear than the offline setting.

**Note From Pc:**

if the above contains the word "oral" or "spotlight" please see: "oral" presentation means -> notable-top-5% and "spotlight" means -> notable-top-25%. As stated in our emails, we are disassociating presentation type from AC recommendations

**Summary Of Ac-Reviewer Meeting:**

n/a